# Assessing Transportation Vulnerability to Tsunamis: Utilising Post-event Field Data from the 2011 Tohoku Tsunami, Japan, and the 2015 Illapel Tsunami, Chile

James H. Williams[1], Thomas M. Wilson[1], Nick Horspool[2], Ryan Paulik[3], Liam Wotherspoon[4], Emily M. Lane[5], Matthew W. Hughes[6],

[1]School of Earth and Environment, University of Canterbury, Christchurch, 8041, New Zealand
[2]GNS Science, Lower Hutt, 5040, New Zealand
[3]National Institute of Water and Atmospheric Research, Wellington, New Zealand
[4]Department of Civil and Environmental Engineering, the University of Auckland, Auckland, New Zealand
[5]National Institute of Water and Atmospheric Research, Christchurch, New Zealand
[6]Department of Civil and Natural Resources Engineering, University of Canterbury, Christchurch, 8041, New Zealand

*Correspondence to*: James H. Williams (james.williams@pg.canterbury.ac.nz)

**Abstract.** Transportation infrastructure is crucial to the operation of society, particularly during post-event response and recovery. Transportation assets, such as roads and bridges, can be exposed to tsunami impacts when near the coast. Using fragility functions in an impact assessment identifies potential tsunami effects to inform decisions on potential mitigation strategies. Such functions have not been available for transportation assets exposed to tsunami hazard in the past due to limited empirical datasets. This study provides a suite of observations on the influence of tsunami inundation depth, road use-type, culverts, inundation distance, debris and coastal topography. Fragility functions are developed for roads, considering: inundation depth; road use-type; and coastal topography, and for bridges, considering only inundation depth above deck base height. Fragility functions are developed for roads and bridges through combined survey and remotely sensed data for the 2011 Tohoku Earthquake and Tsunami, Japan and using post-event field survey data from the 2015 Illapel Earthquake and Tsunami, Chile. The fragility functions show a trend of lower tsunami vulnerability (through lower probabilities of reaching or exceeding a given damage level) for road use categories of potentially higher construction standards. Topographic setting is also shown to affect the vulnerability of transportation assets in a tsunami with coastal plains seeing higher initial vulnerability in some instances (e.g. for State Roads with up to 5 m inundation depth), but with coastal valleys (in some locations exceeding 30 m inundation depth) seeing higher asset vulnerability over all. This study represents the first peer-reviewed example of empirical road and bridge fragility functions that consider a range of damage levels. This suite of synthesised functions is applicable to a variety of exposure and attribute types for use in global tsunami impact assessments, to inform resilience and mitigation strategies.

## 1 Introduction

Road networks are critical to the every-day operation of society, and likewise to the response and recovery phases post-tsunami. Access to impacted populations and repair works to other lifelines can be delayed by roads that are damaged or have reduced levels of service (Eguchi et al., 2013; Horspool and Fraser, 2016; Koks et al., 2019; Nakanishi et al., 2014; Williams et al., 2019). Observations from previous international tsunamis have recorded widespread damage and loss of service to transportation assets including from the 2004 Indian Ocean Tsunami and the 2010 Maule Tsunami, Chile (Ballantyne, 2006; Edwards, 2006; Evans and McGhie, 2011; Fritz et al., 2011; Goff et al., 2006; Lin et al., 2019; Palliyaguru and Amaratunga, 2008; Paulik et al., 2019; Scawthorn et al., 2006; Tang et al., 2006). Defining road asset vulnerability to tsunamis is important for impact assessment and evaluation of mitigation strategies to reduce potential impacts on road networks. In order to do this, robust tsunami vulnerability metrics are required.

Current scientific literature has focused on the development of tsunami vulnerability metrics for damage to buildings, (e.g. Aránguiz et al., 2018; Suppasri et al., 2013) which provide a measure damage or loss for a prescribed hazard intensity. There are few comparable examples for tsunami damage to lifelines infrastructure components (e.g. Horspool & Fraser, 2016; Williams et al., 2019). Commonly used metrics include vulnerability and fragility functions which are used to define the relationship between asset impact level and a hazard intensity (e.g. tsunami inundation depth; Koshimura et al., 2009). Vulnerability functions define the probability of losses (e.g. economic losses) for the given hazard intensity measure, whereas fragility functions provide the probability of exceeding different limit states (e.g. physical damage) for the given hazard intensity measure (Lagomarsino and Cattari, 2015). Fragility functions typically rely on relatively large samples of empirical or modelled impact data, yet quantitative data for road vulnerability have been unavailable prior to recent tsunami disasters. Fragility functions derived from a single tsunami event means they will be characteristic of local asset and event characteristics. For transportation assets, only bridge structures have been analysed for fragility function development (Kawashima and Buckle, 2013; Koks et al., 2019; Shoji and Moriyama, 2007). These studies applied tsunami inundation depth as the hazard intensity measure (HIM) as it usually has a strong correlation with impact and is relatively easy both to model and to measure post-disaster. However tsunami hazard and impact studies to date are almost unanimous in that no single HIM can fully encapsulate the characteristics of tsunami impacts (Bojorquez et al., 2012; Gehl and D'Ayala, 2015; Macabuag et al., 2017; Sousa et al., 2014).

Although post-event tsunami surveys commonly record road impacts as physical damage levels, levels of service can also be considered, which include, but are not limited to physical damage. Coastal road networks are most commonly damaged, or totally destroyed, either by debris impact or erosion of the substrate material (Eguchi et al., 2013; Horspool and Fraser, 2016; Kawashima and Buckle, 2013; Kazama and Noda, 2012; MLIT, 2012), and have reduced levels of service due to debris litter (Evans and McGhie, 2011). Debris litter is a widely identified post-event impact that affects the functionality of an otherwise undamaged road. Prasetya et al. (2012) and Naito et al. (2014) modelled debris transport pathways and debris impact zone potential respectively, with the first noting that debris further inland results in the greatest disruption to lifelines. Neither study

assessed debris density probability for tsunami. Evans & McGhie (2011) notes a correlation between debris sizes as a function of inundation depth to measure spatial distribution, however deposition was not assessed.

Tsunami damage cannot be fully characterised by any one HIM. The topographic setting can also potentially be used to define variations in tsunami damage characteristics. When a tsunami wave reaches the coast, it will travel either long distances inland, at relatively low inundation depths over planar topography, or if confined near the coast will reach considerably greater inundation depths. Planar topography will result in lower retreating inundation speeds whilst the opposite is likely for areas of steep coastal topography (Naito et al., 2014; De Risi et al., 2017; Suppasri et al., 2013).

The objectives of the current study are to (a) analyse post event tsunami survey data to identify potential characteristics of tsunami impacts on road network assets and (b) to develop a suite of tsunami fragility functions for transportation infrastructure asset. This study analyses road asset damage data from two recent tsunamis, the 2011 Tohoku Earthquake and Tsunami, Japan, and the 2015 Illapel Earthquake and Tsunami, Chile. This addresses the gap in global knowledge of tsunami impacts on

transportation infrastructure and ultimately informs tsunami risk reduction. Other than the economic and strategic value that transportation networks provide in post-event response and recovery, transportation assets were selected for the focus of this paper due to them having the only consistently available asset data between the two events. This is in part due to the willingness of organisations to share their network damage data and due to the readily observed assets in-field, which are not obvious for the likes of buried infrastructure (e.g. pipes and cables). The data are analysed considering a range of novel hazard intensity

proxies (e.g. distance from coast) to encapsulate a wider range of HIM's. This better represents road vulnerability to tsunami impacts than using a measure of inundation depth alone.

The Mw 9.0 Tohoku earthquake in the Pacific Ocean, east of Japan (Figure 1b), caused tsunami waves exceeding 30 m inundation depth, in some extreme cases, and affecting much of Japan's eastern coast which was also earthquake-affected

(MLIT, 2012). Transportation infrastructure were extensively damaged throughout the exposed region during this event (Eguchi et al., 2013; Graf et al., 2014; MLIT, 2012). The Illapel event took place on September 16, 2015 in northern-central Chile, triggered by a Mw 8.3 earthquake of the coast of the Talinay Peninsula, (Figure 1a), (Aránguiz et al., 2016, 2018; Contreras-López et al., 2016; Izquierdo et al., 2018; Ye et al., 2015). This event caused localised inundation of up to 7 m, with severe impacts to the transportation infrastructure, the greatest of which were in Coquimbo. While the Tohoku dataset

represents the largest tsunami damage survey for roads, the Illapel dataset represents the first known census style survey of roads impacted by tsunamis. All exposed assets in the Illapel study area (Figure 1a) were surveyed, not only those with observed damage, which was not the case following the Tohoku event.

The Tohoku data analysed in this study were obtained during field surveys and compiled by the Ministry of Land Infrastructure,

Transportation and Tourism (MLIT), whereas the authors collected the Illapel data (Sub-section 2.1). Road and bridge damage and tsunami inundation depth were used to derive vulnerability functions using least square regression and lognormal

probability density functions. The tsunami inundation depth for each asset was obtained by remotely assigning interpolated depth values from the respective surveys (Sub-section 2.1). The data were first analysed for all assets (mixed construction; Sub-section 2.2), then split between use type (as a proxy for construction material; Sub-section 2.2.1), distance from the coast (as a proxy for inundation energy, Sub-section 2.2.2), and distance from the inland extent of inundation (Sub-section 2.2.3), coastal topography (Sub-section 2.2.4), to capture and identify potential variations in asset damage and service levels. Although each analysis gives insight into the broader picture of tsunami impacts on transportation assets, not all data were applicable to the development of fragility functions.

The following sections present the two event datasets, noting a range of hazard-impact trends and observations in the data, which are supplemented with remotely sensed asset and hazard data (Section 2). This includes any trends in the data for topographic setting and asset use type. The results (Section 3) of the analysis are then presented as a suite of vulnerability functions for each applicable hazard-intensity and asset-type combination. A discussion (Section 4) on the results is then presented, which includes their limitations, potential applications and recommendations for future studies, followed by conclusions of the study.

## 2 Methodology

### 2.1 Data Collection

#### 2.1.1 Event 1: Tohoku Earthquake and Tsunami

The Tohoku Earthquake Tsunami provided post-disaster survey teams with an extensive area from which to collect damage data on infrastructure assets. The data used for this analysis are the results of a comprehensive ground survey carried out in the days to weeks following the tsunami by the Japanese Government, City Bureau of the Ministry of Land, Infrastructure, Transport and Tourism (MLIT, 2013). The data relevant to this analysis included detailed road asset damage summaries and local maximum tsunami inundation depths for the exposed area within Miyagi and Iwate Prefectures, which were two of the regions most impacted (Eguchi et al. 2013; Horspool & Fraser 2015; MLIT 2012; Kazama & Noda 2012; Figure 1). MLIT defined the length of affected roads and assigned each section a damage level (Table 1). Much of the study area experienced high levels of long-duration shaking, so not all of the observed damage is necessarily exclusive to tsunami processes (Shoji and Nakamura, 2014). However, it is widely reported in literature and through eyewitness accounts that, in most cases, damage to tsunami-exposed assets were more characteristic of tsunami impacts than with ground shaking. Despite this, some assets would have been damaged, or completely destroyed, by initial earthquake shaking and this co-seismic damage is not recorded in the survey data. Areas with flat topography are not typically consistent with direct road damage from shaking alone. However, where soil liquefaction occurred then this could have resulted in damage, which is not accounted for in this study. The inundation depth and asset data, containing the damage observations, were requested by, and presented to, GNS Science

as GIS shapefiles (.shp). The asset data were presented as edges (lines) representing the true length of each damage observation recorded. The damage data were supplemented with a Japanese-to-English translated spreadsheet of instructions and explanations. Modelled maximum inundation depth (m) was available in 100 x 100 m grid cells, across the study area (Eguchi et al. 2013; MLIT 2012; Horspool & Fraser 2015). This empirical dataset is one of only few in existence globally for transportation damaged by tsunami, which is why it is included for this study.

### 2.1.2 Event 2: Illapel Earthquake and Tsunami

A census-style field survey was conducted in Coquimbo, Chile, between 8-12 days after the Illapel event by a New Zealand-based team of five. The New Zealand Society for Earthquake Engineering requested and accepted an invitation from the Chilean Association of Seismology and Earthquake Engineering to undertake a collaborative field survey. The team included members from GNS Science, National Institute of Water and Atmospheric Research (NIWA), Wellington Lifelines Group, Auckland City Council, University of Canterbury and was supported by Chilean researchers from University of Valparaiso. Coquimbo was selected as the focus of the post-event survey as it was the region most impacted in this event and also represented a small enough study area to collect data in a short timeframe. Damage, asset and hazard data were collected, using the Real-time Individual Asset Collection Tool (RiACT) in accordance with International Tsunami Survey Team (ITST) procedures (Lin et al., 2019). Observations were recorded as points, and in the case of roads, a point was placed in the centre of each observation with a length of observed damage also recorded, among other attributes.

The survey area experienced low peak ground accelerations (0.20g - 0.29g, USGS, 2015) in this earthquake event and subsequently road damage can be assumed as only tsunami induced. This assumption was corroborated through informal discussion between the field survey team and members of the public. Road damage was defined using a four-tier damage-level classification in accordance with the MLIT classification structure (Table 1). This was done to be consistent with the Tohoku dataset, which was already available and represented the largest damage repository of tsunami impacts on roads and bridges, as outlined in the subsection below. Although this classification of damage level could have been refined, the field team decided it still represented a relatively efficient method in-field and at a resolution high enough to incorporate the range of observed damage types. Most roads in the inundation area were founded on sandy material, with a compacted granular subbase and a thin asphalt surface (flexible pavement construction method; M E Nunn, A Brown, 1997; NZTA, 2014). There were few 'both-lane' wash-outs, with minor / single lane wash-outs being more common, and many washouts occurred where a culvert ran beneath the road surface. Inundation depth indicators (watermarks) were also collected in the field by measuring watermarks against vertical structures (e.g. buildings, utility poles). A total of 978 watermarks were recorded across the survey area which represented maximum inundation depth above ground level. The total survey area included an approximately 7 km stretch of coastline.

## 2.2 Data Analysis and Damage Observations

The first step in defining vulnerability is to develop fragility functions, which require: a spatial hazard metric(s) (HIM);
measured or descriptive spatial asset data (both damaged and non-damaged); asset attribute information. The HIM and asset attribute information are the two key variables when considering vulnerability and both are considered, independently and in tandem, to define vulnerability of assets. The most common HIM is inundation depth, and the first step was to use this data to calculate fragility functions for mixed construction assets. With the Coquimbo dataset, roads were separated into approx. 50 m sections, and assigned the corresponding damage level (DL0-DL3) and inundation depth, through the watermark
interpolation, at the centre of each feature (Figure 2) using inundation depth bins of 0.25 m (0.0 - 0.25 m, 0.25 – 0.5 etc.). The total length of road (in km) for each depth bin and for each damage level was tabulated for each HIM by count and proportion (Figure 3). Once inundation depth had been considered, other HIM were used to define vulnerability more holistically. Fragility functions were then developed, as described in more detail in sub-section 2.3.

The Tohoku dataset lacked spatial non-damaged asset data (DL0), which is crucial to defining proportional damage probabilities. Therefore, all roads within the inundation area were extracted from OpenStreetMap (OSM), (OpenStreetMap contributors, 2015) or were digitized from aerial imagery and those which were not recorded in the MLIT data were assumed undamaged (DL0). This resulted in a complete dataset of roads and bridges exposed to the tsunami, each with an observed damage level (DL0 – DL3). Figure 4 shows an example of observed damage levels for roads in the town of Ishinomaki within
the study area. Tsunami inundation depths MLIT (2012) were then assigned to each road section using 1 m inundation depth bins (i.e. 0.0 – 1.0 m, 1.0 – 2.0 m etc.). Larger inundation depth bins were used compared with the Illapel dataset (i.e. 1m vs 0.25m), as there were greater hazard intensity values (> 10 m vs < 4 m). Road length totals in each hazard intensity bin were tabulated for each damage level) by count and proportion (Figure 5). The results of this analysis are presented in Section 3 as fragility functions.

Asset attribute information should include road construction type, allowing for the development of construction specific fragility functions. As this was not included in the MLIT, 2012, dataset, the closest equivalent was road-use type category based on jurisdiction (0; Unclassified, 1; State road, 2; Main local road, 3; General prefectural road, 4; Municipalities road, 5; Lowest class road). These classifications were then converted to road-use type equivalent categories (0; Unclassified, 1;
Motorway, Trunk, Primary, 2; Secondary, 3; Residential, Road, 4; Tertiary, 5; Construction, Service, Unsurfaced) to ensure compatibility with OSM (OpenStreetMap contributors, 2015) data (DL0). Roads digitised from satellite imagery were assumed to be in class 3. However, those that could not be classified were 'Unclassified' (0) which have not contributed towards the resulting fragility functions. These road use classes link to different traffic loading levels, which inform road design, therefore these data broadly encompass differences in construction type, but some degree of overlap is assumed.

The analysis used for Tohoku bridge vulnerability was similar to that of roads, however inundation depth was normalised to the height above the base of a bridge deck. OSM data already included bridges as a separate road attribute and so were easily integrated, and satellite imagery was used only to validate that all bridges were included. Bridge construction materials were not available, neither was bridge deck base height above ground, both of which would be necessary for a higher resolution fragility function (Horspool & Fraser 2015; Shoji & Moriyama 2007).

The MLIT dataset had an assigned bridge damage level between DL1 & DL3 (Table 1). All non-surveyed bridges in the inundation area were assumed to be undamaged and consequently assigned DL0. Figure 6 shows an example of observed damage levels for bridges in the town of Ishinomaki within the study area. Modelled tsunami inundation depth was assigned at the centre point of each bridge to avoid a bridge falling within multiple inundation depth bins. Since deck base height was not included in the dataset and in many cases the hazard layer did not include depths within river channels, to estimate the inundation depth above deck base height the inundation depth at the bridge abutment was used and the assumption made that in most cases the deck would be relatively level with the abutment, although the deck height (thickness of beams and roadway) is still not considered. Bridges in each hazard intensity bin were tabulated for each damage level by count and proportion (Figure 5) and resulting fragility functions are presented in Section 3.

### 2.2.1 Culverts Associated with Increased Road Impacts

While inundation depth has been used as the HIM, as outlined above, other potential metrics that might have a bearing on asset vulnerability were also considered. As mentioned in Sub-section 2.1, in Coquimbo, Chile, road damage was observed at many culvert openings especially along the coastal esplanade (Figure 2). This damage is consistent with the principle of contraction scour (Duc and Rodi, 2008), which occurs when the depth of inundation exceeds an opening and the inundation becomes contracted. The inundation is directed down and through the structure, causing an increase in the velocity and shear stress around the outlet, therefore increasing scour. Inundation speed, inundation depth, the degree of submersion and size of the culvert are all factors dictating contraction scour intensity. Scour can also be exacerbated by the enhanced turbulence and vortex formation in this inundation. Scour around culverts can also be caused by the back-inundation after a tsunami has receded. Recorded culvert locations were used to assign the presence of a culvert/outfall pipe (present '1', not present '0') to each 50m section of road. The frequency and proportion of road sections with a culvert were tabulated for each damage level (Figure 7). DL0 had a count of 573 road sections (too many to represent in Figure 7a), with five having a culvert. This analysis is not conducive with fragility functions, due to the limited number of culverts surveyed, so none are developed in this study.

### 2.2.2 Distance from Coastline

Tsunami inundation velocity is known to have a considerable influence on asset impacts, especially due to scour. However, inundation velocity data were not available for the Illapel dataset, so distance from the coast was used as a proxy for inundation velocity. This assumes a constant deterioration of landward wave energy including horizontal and vertical buoyancy pressure as a tsunami wave moves inland (from friction and gravity). This was observed for road assets in Coquimbo as damage levels

reduced with distance from the coast. A measure of distance from the coast was calculated at 25 m inundation distance bins (i.e. 0.0 – 25.0 m, 25 – 50 m etc.). Each road section was assigned the associated distance from coastline value and the results were tabulated for each damage level as counts and proportions of damage (Figure 8). Since distance from the coastline is not a direct damage causing process, the analysis not conducive with fragility functions, so none are developed for this study.

### 2.2.3 Debris Based Level of Service

Another consideration of vulnerability is to look at various impact types. As mentioned in Section 1, debris can cause considerable disruption to transportation networks, through direct damage and through blocking routes. Therefore the effects of debris on an asset's level of service is considered, and a new HIM (distance from the landward inundation extent) is used. To assess the correlation with debris and a roads level of service in Coquimbo, debris distribution data is required. However debris clean-up had begun prior to the survey, so publicly available drone mounted camera footage (Puerto Creativo, 2015, 2016) was used to map out debris density on roadways. These were classified into five service levels (SL), as defined in Table 2. SLU represents areas of ponding observed and is classed separately since the depth and amount of debris entrained is not known. If a road was not associated with debris deposition, it was assigned SL0. To account for potential horizontal sorting of debris, the distance from tsunami inundation extent (i.e. the greatest recorded landward observations of tsunami inundation) was used and each road was assigned an associated value. The tsunami exposed area in Coquimbo was predominantly flat topography, with only a few instances of a retaining wall or incline bounding the landward inundation extent. The local sea port, of which are typically well-defined regions of debris origin (Naito et al., 2014), was located along the South-West to North-West inundated coastline.

As well as inundation depth (m) and distance from the coast (m), each road section was now assigned a level of service (SL0 - SL3 or SLU), (Figure 9), and a distance from the inundation extent value (in m). For each distance measure, road length frequency was tabulated by 25 m bins (i.e. 0.0 – 25.0 m, 25.0 – 50 m etc.), for each service level (Figure 10). There was no such empirical source of debris density observations available for the 2011 Tohoku Tsunami, so this is not considered in the analysis of the Tohoku dataset.

### 2.2.4 Coastal Topography

Fragility functions that do not consider topography may not accurately represent tsunami damage characteristics when used for subsequent impact assessment. Therefore this study defines vulnerability for two broad coastal settings, 'coastal plains' and 'coastal valleys', to develop specific vulnerability curves similarly to De Risi et al. (2017) and Suppasri et al. (2013). The data for Tohoku roads, presented above, were further refined by assigning each road section an applicable topographic setting (Figure 11). For the two different topographic settings, the number and proportion of road sections in each damage level were tabulated against inundation depth (Figure 12). The resulting fragility functions are presented in Section 3.

## 2.4 Developing Fragility Functions

The asset damage probabilities for each damage level were calculated and shown against a median value within increasing HIM bins, to account for lower amounts of data at higher HIM. Following the methods of (Koshimura et al., 2009) linear regression analysis was performed to develop the Log-normal cumulative distribution function vulnerability curves. A probability $P$ of reaching or exceeding a damage level for a given hazard intensity value is given by either Eq. (1) or (2):

$$P(x) = \Phi\left[\frac{x - \mu}{\sigma}\right] \tag{1}$$

$$P(x) = \Phi\left[\frac{\ln x - \mu'}{\sigma'}\right] \tag{2}$$

where $\Phi$ is the standardized normal (lognormal) distribution function, x is the HIM (i.e.. inundation depth), $\mu$ and $\sigma$ ($\mu'$ and $\sigma'$) are the mean and standard deviation of x (ln x) respectively. Two statistical parameters of fragility function, i.e. $\mu$ and $\sigma$

($\mu'$ and $\sigma'$), are obtained by plotting x (ln x) and the inverse of $\Phi^{-1}$ on normal or lognormal plots, and performing the least-squares fitting of this plot. Two parameters are obtained by taking the intercept (= $\mu$ or $\mu'$) and the slope (= $\sigma$ or $\sigma'$) in either Equation (3) or (4), depending on the result of the least-squares fitting:

$$x = \sigma\Phi^{-1} + \mu \tag{3}$$

$$\ln x = \sigma'\Phi^{-1} + \mu' \tag{4}$$

The resulting fragility functions from each dataset are presented in the following section, although not all of the data analysed in this study are applicable to the development of fragility functions.

## 3 Results

Variations in asset impacts are presented with the developed fragility functions each reflecting a potential difference in damage probability due to (1) damage level only (for each event data), (2) road use type, (3) distance from coastline (as a proxy for inundation velocity), (4) debris based level of service, (5) topographic setting and (6) a consideration of both use type and topographic setting. The results of this study are presented in Table 3 and the following sub-sections.

### 3.1 Damage Level

The exposed roads assessed in this study in general perform well, even under the highest inundation depths. There is less than 0.2 and less than 0.3 probability of complete washout (DL3) at 15 m inundation depth for roads and bridges, respectively, in the Tohoku dataset (Figure 13a and Figure 14). Roads in Coquimbo have less than 0.25 probability of complete damage (DL3) at 15 m (Figure 13b). By comparison, a reinforced concrete building has a 0.4 probability of reaching or exceeding complete

damage at the same inundation depth (Suppasri et al., 2013). All Tohoku road damage levels are at a lower probability than that of the equivalent Tohoku bridges. This is to be expected as each asset has a different tsunami loading regime, with road impacts associated with scour, while bridge impacts are related to horizontal loading across piers and the superstructure as well as vertical loading across the bridge superstructure. Bridges are typically exposed to higher levels of hydrodynamic forces (both horizontal and vertical) as tsunami flows are concentrated in the channels these bridges span. Although not considered in this study, flexible bridge connections will reduce tension from tsunami loadings when compared to rigid (i.e. steel) connections. A higher flexibility in the substructure will also reduce horizontal tsunami loadings (Istrati et al., 2017; Istrati and Buckle, 2014). The Coquimbo roads are at a higher probability of damage compared with the Tohoku roads and bridges. This is to be expected given the differing levels of construction standards in Japan in comparison to Chile. The Illapel study area did not contain any roads that could be considered equivalent in capacity to the likes of Japanese State Roads, which would be given the highest design standards in terms of maximum flexibility and loading design. This will have resulted in a lower overall vulnerability for mixed construction Tohoku roads (Figure 13a) when compared with the mixed construction road vulnerability for Illapel (Figure 13b). None of the functions have a probability of 1.0 within the parameters of the presented results (i.e. up to 15m inundation). This is a reasonable interpretation as roads and bridges, although particularly vulnerable under certain conditions, are far more resistant to tsunami impacts than many other assets (Williams et al., 2019). As a comparison, mixed construction buildings in the 2011 Tohoku Earthquake and Tsunami had a probability of 1.0 at all damage levels when inundation exceeds 10m (Suppasri et al., 2013). Bridge piers and abutments are designed with scour, horizontal loading and vertical loading from moving water in mind, although not specifically for tsunami forces, whereas the foundations and structures of buildings are typically not, making them more vulnerable to tsunamis.

The results of the analysis on culvert locations and the associated road damage levels in Coquimbo indicate a correlation with the presence of a culvert and an increased damage level (Figure 15). This indicates that all instances of a culvert in this event have resulted in road damage to some extent, and in most cases moderate or severe damage (DL 2 and DL3).

### 3.2 Distance from Coastline

The analysis for distance from the coast, as a proxy for inundation energy, did not warrant the development of vulnerability curves (Sub-section 2.2.2). However, the results (Figure 16) show a clear trend between higher probabilities of damage occurring closer to the coastline. This may be an indicator for deteriorating wave energy (due to surface friction and gravity) but could simply be an indicator of increased inundation depths at the coast since there is no empirical evidence of hydrodynamic forces in the Illapel event. The same was noted in a study of building vulnerability in the Illapel event (Aránguiz et al., 2018), particularly with lower damage occurring beyond a wetland area and behind a raised railway ballast, when compared to those nearer the coast.

### 3.3 Debris Based Level of Service

Tsunami debris transport is a function of inundation depth, inundation velocity and debris size resulting in horizontal sorting of objects toward the in-land inundation extent as larger materials fall out of suspension (Charvet et al., 2014; Evans and McGhie, 2011; Naito et al., 2014; Prasetya et al., 2012). In Coquimbo debris-based level of service analysis are indicative of this statement as a higher proportion of roads have debris deposited on them between 0 and 150 m from the landward extent of tsunami inundation (0 – 22% of maximum in-land inundation extent). This indicates debris carried in-land falls out of horizontal suspension prior to reaching the maximum in-land extent. The results from the debris density analysis show that there is higher debris density between approximately 75.0 – 150.0 m (11 – 22%) from in-land inundation extent (Figure 17). Debris density probability is consistently lower, for all levels of service, between 0.0 – 75.0 m (0 – 11%) from in-land inundation extent and 200.0 – 672 m (30 – 100%) from in-land inundation extent. SL1 has a much higher probability of occurrence than SL2 and SL3 at distances > 200 m from in-land inundation extent (Figure 17). This is consistent with previous studies and field observations where debris is consistently distributed across an inundation area during landward and seaward inundations.

### 3.4 Road Use Type

Using the mixed construction road data for exposed areas of the Tohoku event, the different structural types are split into broader use categories, as the closest approximation of construction material, type and method, for the development of fragility functions (Figure 18). The most resistant use category, with respect to tsunami impacts, is State Roads (Figure 18a), with all damage levels being of a lower probability than the other use categories. This is followed by Main Local Roads and General Prefectural Roads (Figure 18b & c), each with very similar probabilities of DL1 and DL3, although Main local roads have higher probability of reaching or exceeding DL3 at less than 7 m inundation depth. Main Local (Figure 18b) roads also have a higher probability of reaching or exceeding DL2 with comparison to General Prefectural Roads (Figure 18c). This can be interpreted as Main Local Roads having a certain characteristic that pushes them from DL1 to DL2 much faster than with General Prefectural Roads. It is likely these two classes share similar construction standards and materials. Municipalities Roads (Figure 18d) have considerably higher probabilities of reaching or exceeding each DL, with a much steeper gradient when compared to road class 1, 2 & 3 (Figure 18a, b, & c). The most vulnerable roads are the Lowest Class Roads (Figure 18e), which we cautiously assume here are unsealed based on pre-event satellite imagery. Given these roads are scarce in the mostly urban environment of the study area and the unknown nature of their construction, the data for this road class were not sufficient to classify DL1 and DL2. At 3 m of inundation depth, this road class already exceeds a probability of 0.5 of complete damage (DL3).

### 3.5 Coastal Topographic Setting

The previous fragility functions for the Tohoku event, presented above, represent an average of the data for the whole of the tsunami-exposed area. This sub-section presents the results of the analysis looking at the effects of two different topographic settings on tsunami damage to roads. An example of the difference in these coastal topographic settings is that at 2 m of inundation depth there is ~ 0.09 probability of DL3 on plains, whereas only ~ 0.05 in valleys (Figure 19a & b). The damage probability in plains increases to ~ 0.11 at 10 m inundation depth, while the damage probability is ~0.16 for valleys. It is noted

that the damage probability for the plains abruptly increases from 0 to 0.08 (at around 0.03 m) while for valleys 0.08 is not reached until 3 m.

Since it was already established in Sections 1 and 2, road use type (as an estimation of construction type and material) is an important factor in defining tsunami vulnerability. Therefore, the Tohoku data is again split into different road classes to

355 compare, in more detail, the effects of coastal topography on tsunami vulnerability. Two examples are presented below, for State Roads and Main Local roads (Figure 20). Road classes 3 – 5 are not presented here, as these did not have sufficient data to warrant fragility functions. In general, the damage probabilities for roads in the valleys are higher than those on plains for each inundation depth. However, the $r^2$ values for coastal valleys are particularly low, so the comparisons between each coastal setting my not be entirely representative of true vulnerability

### 4 Discussion

This study represents the first attempt at developing empirical tsunami fragility functions for roads. Although previous studies have developed fragility functions for tsunami impacts on road bridges using the Tohoku dataset (Eguchi et al., 2013), they do not include undamaged assets in the analysis. This is a considerable drawback given the number of undamaged assets is equally important in developing cumulative distribution functions for damage probability. The fragility functions presented in this

study, particularly those based on Tohoku data, have a number of potential applications within a broader risk reduction framework, particularly in developed countries with similar construction standards to Japan. These can be used in impact and loss forecasting to provide high resolution estimates (i.e. considering topographic setting and construction material/type), or for more rapid loss modelling if implementing mixed construction and topographic setting curves. The number of refined curves presented in this study provides flexibility for future global applications. Applications in countries that do not share

similar construction standards to Japan or Chile are still possible, ensuring a full understanding of the limitations, for example, a country with different levels of construction standards may have a number of exposed roads that share similar construction standards to a class 4 Japanese road.

This study represents the first empirical analysis directly linking the presence of a road culvert with increased probability of road damage. Although this analysis did not warrant the development of fragility functions, given an applicable case study and

375 consideration of the limitations, these results could be used for weighted tsunami impact assessment. The evidence from this

study certainly indicates a need for the consideration and development of mitigation strategies to reduce the associated vulnerability of transportation assets located adjacent to culverts exposed to tsunamis.

The analysis of debris deposition density in Coquimbo, and the associated level of service to roads, is also the first of its kind. As with culvert damage, this dataset did not warrant the development of fragility functions, however under the right conditions it could be applied to a weighted vulnerability metric within a wider tsunami impact assessment of road infrastructure. The analysis methods can also be applied to future events.

This study highlights that the collection of post-event tsunami impact data is invaluable for vulnerability analysis of infrastructure assets, which have been under-represented in past studies. The methods used for data collection in this study show that a combination of empirical field survey data and post-survey remote sensing could be an effective way to supplement and refine field observations. In the case of Illapel, the survey was conducted using only a measuring tape and observations recorded on a tablet. This demonstrates that relatively simple survey techniques and equipment can be used to provide rapid 'in-and-out' surveys after events of this magnitude, in order to collect data on assets that would otherwise not be included by other survey teams.

## 4.1 Limitations

The Illapel data is from a relatively small tsunami event in a localised area of one single coastline. This represents a small sample size but a high level of detail applied though a census style survey. The Tohoku data is from a considerably larger sample size but asset characteristics provided to the authors of this study are in some cases recorded at a low resolution (e.g. the lack of recorded construction material). The quantity of data is important when developing vulnerability curves for a dataset of the Tohoku scale. For example, fragility functions for road class 3-5 on the two coastal topographies would not have been applicably comparable to that of class 1 & 2 roads due to the different quantities of data, and particularly at higher inundation depths. Whereas the more localised data of Illapel has more consistent quantities of data across the range of inundation depths but is also limited by the overall data size. For example, the dataset did not warrant the comparison of different coastal settings since only flat topography was represented in the study area, which is also noted by Aránguiz et al., 2018 in the context of building vulnerability. The Tohoku survey also did not include undamaged (DL0) roads, which as outlined in Section 2, were remotely sensed through this study on the assumption that any roads not included in the data were DL0. This assumption depends on a range of factors including scope of survey, access, classification criteria and the accuracy of shapefiles used to classify DL0. DL0 is more likely over represented in this study than under-represented.

The Tohoku data represents not only the effects of tsunami hazards but also from seismically induced shaking, including soil instability. Although in most cases the observed damage will be characteristic of tsunami impacts, some assets may have been initially weakened by seismically induced hazards, including shaking, liquefaction, landslides, lateral spreading and differential settlement. This can be interpreted as the fragility functions potentially over-estimating the actual vulnerability of Japanese transportation assets to tsunamis.

As with any field survey data, there is inherent bias in each individual surveyor's assignment of damage levels. With respect to the Illapel data collection this was controlled to some extent by using one consistent survey team member for each asset recorded. It is more difficult to evaluate the Tohoku dataset in this regard, but it is reasonable to assume the MLIT survey team grappled with and attempted to mitigate similar issues. However, it is reasonable to note that a subjective bias is possible with both datasets, particularly when comparing with equivalent damage classifications in other tsunami events. The methods used to spatially define the Tohoku dataset (i.e. 50m sections of road) are not as applicable with smaller datasets, such as with the Illapel event. Since some of the DL3 road washouts were smaller than 50m in Coquimbo, there is potentially an over-representation of this damage level, and potentially even DL1 &DL2. This means the Illapel curves may be an over estimation in terms of damage. In addition, some of the worst damaged roads had already begun repair works and may have been over represented in the survey.

The results of this study are also limited, to a certain extent, by the methodology used to fit the impact data to fragility curves. Some curves overlap at the lower hazard intensities (e.g. Figure 14), since the data is treated nominally when fitting curves. This could be addressed by adopting a cumulative link model which fit raw asset impact and hazard intensity data to fragility curves simultaneously (Lallemant et al., 2015).

Information on Japanese culvert locations was not available to the authors of this study. It would have been useful to validate the positive relationship between culverts and road damage in a tsunami, against that identified in the Coquimbo case study. We note this may be a fruitful future study. The results for increased road vulnerability associated with the presence of a culvert may be under-represented. The field survey was thorough in its collection of data, however, if culverts/outfall pipes were covered with debris or sediment they will not have been recorded. Similarly, regarding the classifications for levels of service for debris affected roads in Coquimbo, the drone footage used covered approximately 90% of the inundation zone. Therefore, some areas may be under-represented for debris deposition as a result.

Another limitation of using another team's survey data is the assumptions made around asset classifications. In the road use-type fragility functions, road class 1 shows very low vulnerability to inundation depth. These would include Japan's most highly engineered road assets implying higher construction standards compared with other road use classes. Road classes 2, 3 and 4 trend very similarly and likely share a similar spread of road construction standards. These also show considerably lower vulnerability to inundation depth than that of class 5 roads. Class 5 roads show high vulnerability at even low inundation depths. This class likely includes roads highly susceptible to erosion. This suggests that at the resolution of this road classification method, there is potentially only a need for three broader classes – highly engineered, standard and low-grade. However ideally the various range of road types considered in this range of data would be separated, which is outside the scope of the present research. The subtleties of use-type classification vulnerability may be a function of geophysical setting of each road class, which could not be tested due to a lack of available high-resolution soil data.

## 4.2 Future Research

This study presents a full analysis of empirical post-event tsunami impact data from field survey through to refined fragility functions. It can therefore be used as a framework for similar analysis of transportation impact data from future tsunami events. Data from future tsunamis may also provide some degree of refinement to the results presented in this study. Future work should also consider the data collection and analysis of a range of other critical infrastructure assets, such as electricity, telecommunications, water and fuel. There is a considerable knowledge gap on tsunami impacts on infrastructure, which should be better addressed to inform risk reduction strategies.

The Tohoku dataset of tsunami-damaged roads remains the most extensive in the world and the vulnerability curves developed from them in this study could be even further refined through more complete data. One particular limitation addressed above is the high concentration of co-seismic hazards the roads were exposed to prior to tsunami inundation. It would be possible to eliminate some assets which were likely earthquake-damaged by using high resolution geomorphic, soil and liquefaction hazard data from 2011, of which were not available to the authors of this study. Given the results of the culvert analysis, future post-event survey data can be used to corroborate the increased vulnerability of roads associated with culverts. This can also be used to inform potential mitigation strategies to increase the resilience of roads and culverts alike. This could also be done for the Tohoku dataset if pre-event surface drainage channel data (i.e. an indicator of culverts location) were used, of which was not available to the authors of this study. Similarly, the Tohoku dataset could be further refined by eliminating potentially undamaged roads (those covered in debris during the survey but not physically damaged). Aerial or satellite imagery could be used for this or using the observations from the Illapel debris analysis (Figure 17) in this study to apply a proportional alteration to the dataset (e.g. removing an equivalent proportion of roads within 11 – 22 % of the inundation extent).

Future post-event tsunami surveys should include data on debris dispersal and deposition if possible. It is acknowledged that a lack of time and resources often plays a defining role in the type and quantity of data collected by survey teams, but if technology such as high definition (HD) drone footage or rapid HD aerial photography were conducted after tsunami events, then this could be combined with ground-level observations on debris. This is often not possible as communities begin cleaning debris almost immediately after an event. In the case of Coquimbo, all roads were cleared of debris by the field team's arrival, 6 days after the event.

During the analysis, several interesting damage characteristics were potentially identified, although there were not enough data to develop robust fragility functions, particularly as the small sample size at Coquimbo reduces the ability to derive a robust statistical sample. Therefore, the observation remains qualitative and the parameters require further investigation from future events. One such observation being the potentially increased chance of road damage in Coquimbo given the presence of a culvert (91% roads with culverts $\geq$ DL2). Future post-event tsunami surveys should consider the collection of data for these types of observations to validate against those presented in this study.

## 5 Conclusions

Data from two comparable tsunamis are used to develop fragility functions for roads and bridges. The results of this analysis conclude that:

- Roads with higher construction standards perform better during tsunamis than those with lower standards. This is evident in use-types (based on design parameters based on capacity), showing the higher capacity roads have lower tsunami vulnerability.

- Bridges are more vulnerable to the impacts of tsunamis than roads. However, a more appropriate direct comparison is between buildings and bridges, of which bridges are better designed to withstand the forces of tsunami loading and have a lower level of vulnerability at all hazard intensities (Inundation depth) compared with buildings.

- Field survey observations can be effectively supplemented with remotely sensed data to compare various HIM with subtleties in asset attributes to define tsunami vulnerability including:
  - Roads in coastal valleys are more vulnerable than those on coastal plains, however 'State Roads' on coastal plains have higher vulnerability at low inundation depths, compared to coastal valleys, which is then exceeded at higher inundation depths in coastal valleys, when compared to coastal plains.

  - Culverts represent particularly vulnerable sections of roads due to the effects of contraction forces on the associated subgrade they are embedded through.

  - Debris are horizontally sorted across areas of tsunami inundation with the highest densities of deposition found within 75 and 150m ($11 - 22\%$) from the inland extent of inundation (in the case of the Illapel event). Greater densities of debris on a road decrease its level of service.

The suite of tsunami fragility functions for transportation assets presented in this study address a considerable gap in global knowledge. These functions can be applied through tsunami impact assessments to inform tsunami risk reduction strategies. Future tsunami impact surveys should collect more data, especially on infrastructure asset attributes, at higher spatial resolutions, and rapid post-event data capture is critical to the development of robust fragility functions.

**Author contributions.** JW, NH and RP conducted fieldwork. All authors contributed to the manuscript preparation (JW, TW, NH, RP, LW, EL and MH).

**Acknowledgements.** The authors would like to acknowledge the Ministry of Land Infrastructure Transport and Tourism
(MLIT) for the provision of field-survey data, as well as funding contributions from GNS Science; the National Institute of Water and Atmospheric Research (NIWA Taihoro Nukurangi) project code CARH2006; Resilience to Natures Challenges (MBIE) Resilience Rural Backbone (GNS-Resilience003); Earthquake Commission capacity fund to Geological Sciences,

University of Canterbury; The University of Auckland; the Mason Trust Fund; Environment Canterbury; and the Christchurch City Council.

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

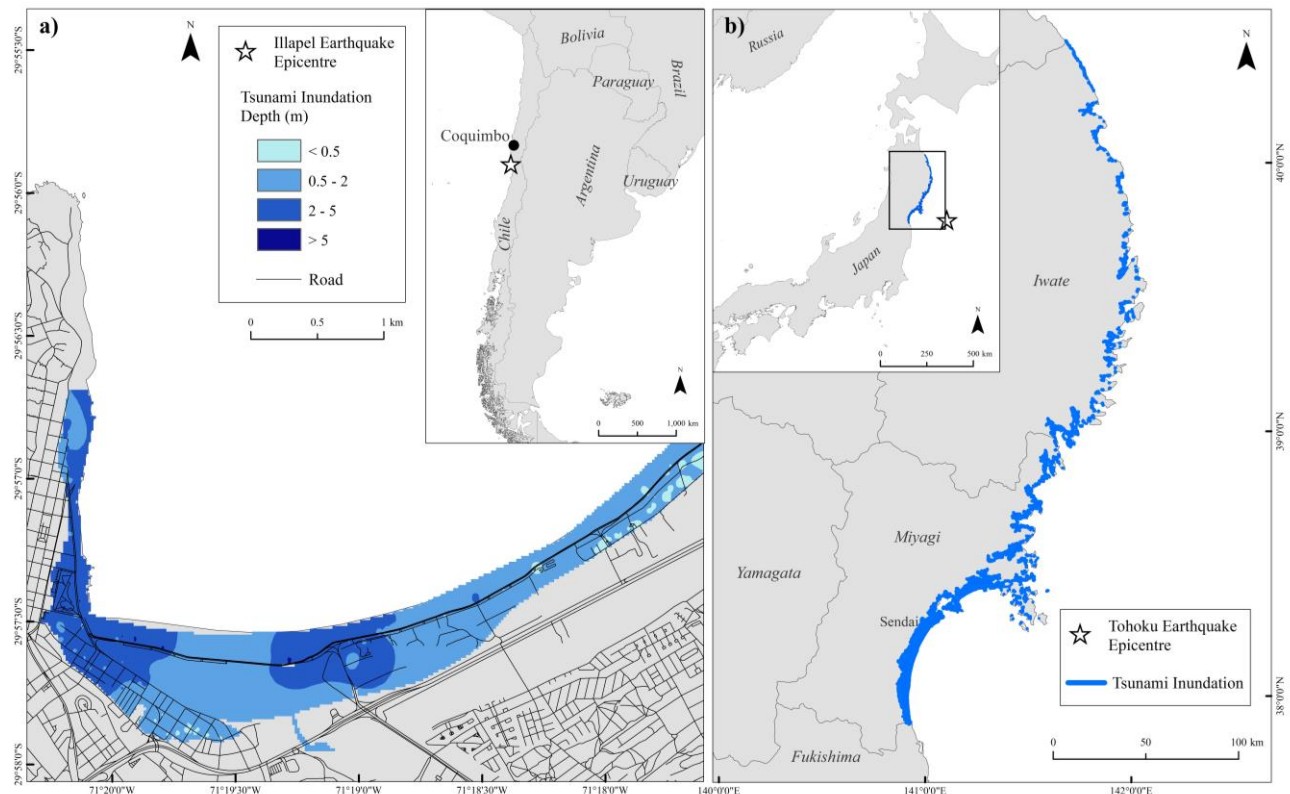

**Figure 1: Tsunami inundation in Coquimbo for the 2015 Illapel tsunami, Chile (a), and in Miyagi and Iwate Prefectures for the 2011 Tohoku Tsunami, Japan (b), © OpenStreetMap contributors 2015. Distributed under a Creative Commons BY-SA License.**

**Table 1: MLIT damage classifications for roads and bridges (MLIT, 2012) and field examples of road damage levels from the 2015 Illapel Earthquake and Tsunami, Coquimbo, Chile, and equivalent bridge examples from the 2018 Sulawesi Earthquake and Tsunami, Indonesia, respectively**

| Damage Level | 0 | 1 | 2 | 3 |
|---|---|---|---|---|
| Damage State | No Damage | Minor | Moderate | Severe |
| Road Damage Description | *No damage* | *Minor damage to road surface. All lanes passable* | *Major damage to one lane. One lane impassable* | *Major damage to whole carriageway. All lanes impassable* |
| Road Image | | | | |
| Bridge Damage Description | *No damage* | *Minor damage, often from impacts to the superstructure* | *Major damage to superstructure but still in place on piers. Superstructure may have been shifted* | *Complete washout of superstructure* |

| **Bridge Image** | 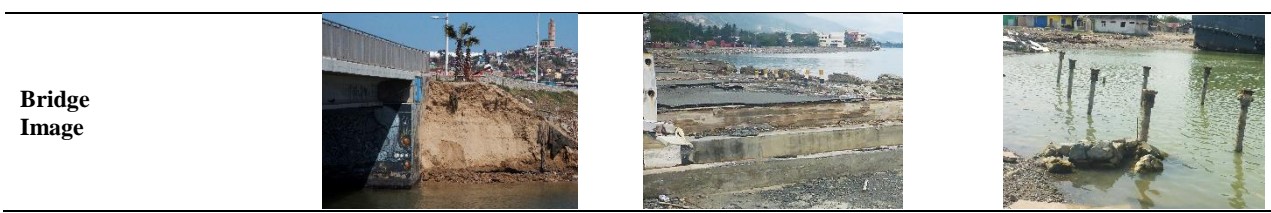 |

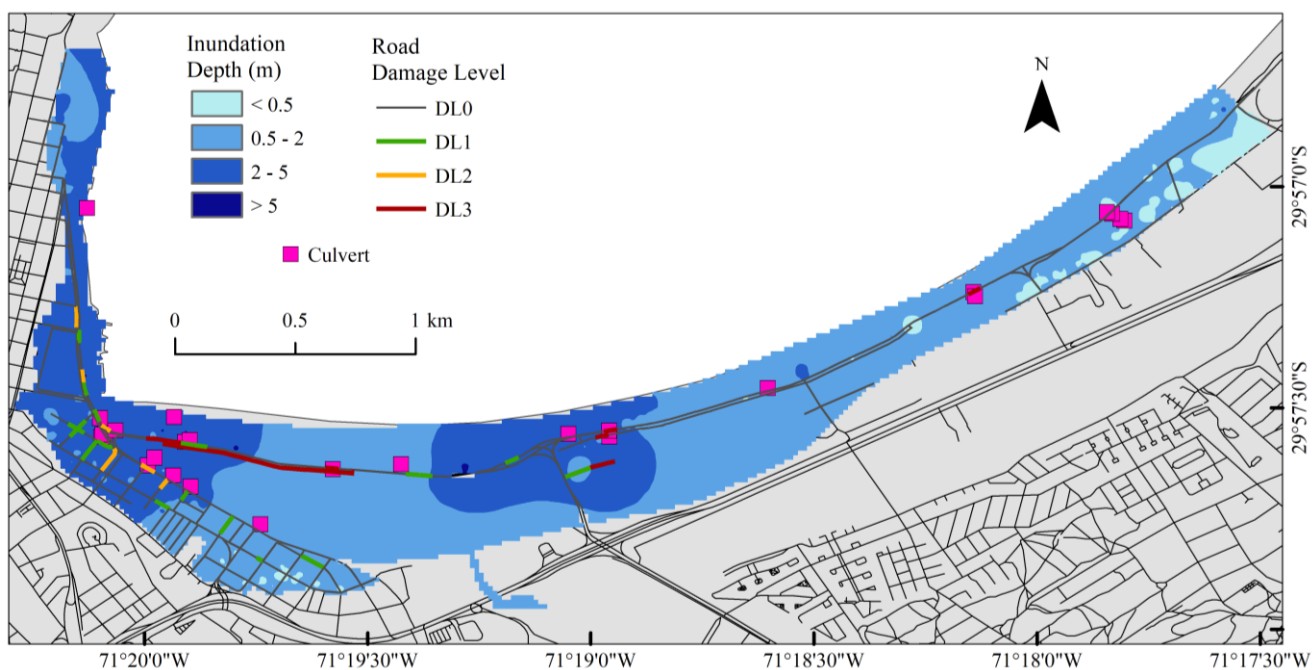

**Figure 2: Tsunami inundation, road damage level and culvert locations in Coquimbo, Chile following the 2015 Illapel Earthquake and Tsunami, © OpenStreetMap contributors 2015. Distributed under a Creative Commons BY-SA License.**

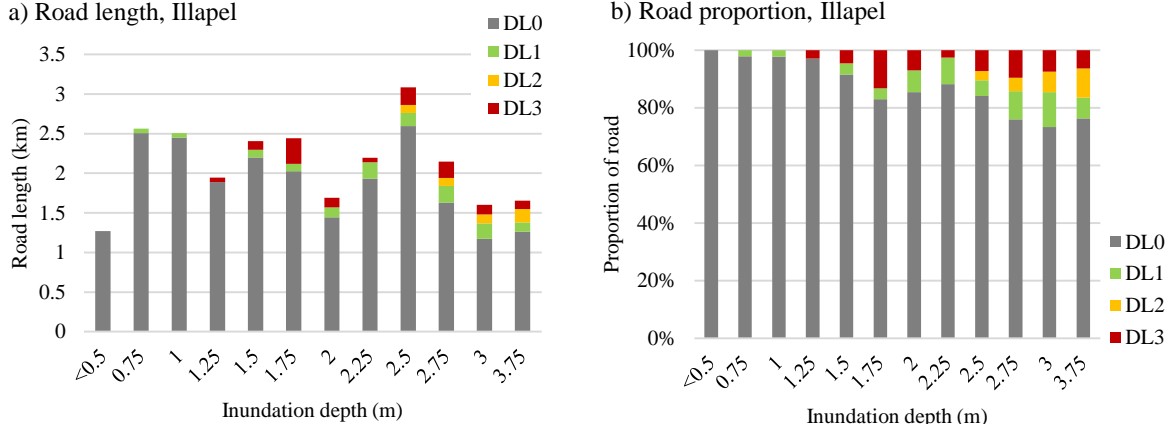

**Figure 3: Total length (a) and proportion (b) of exposed roads, by inundation depth, for the 2015 Illapel Earthquake and Tsunami**

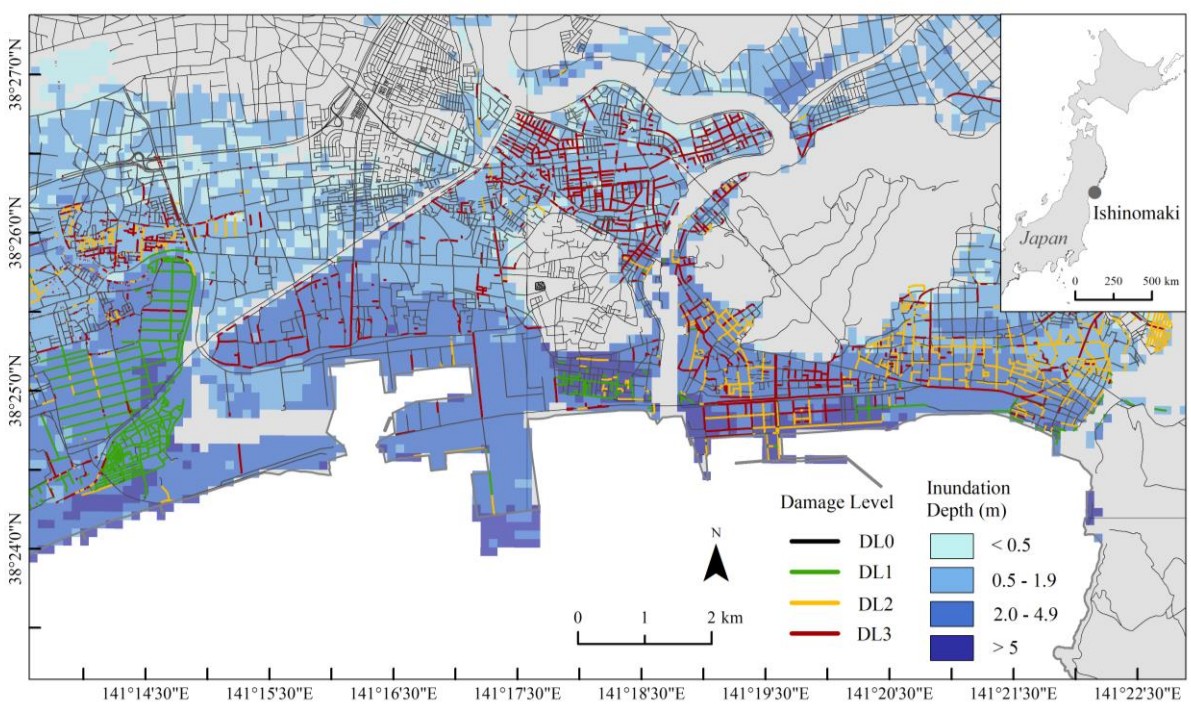

**Figure 4: Tsunami inundation and road damage in Ishinomaki, from the 2011 Tohoku Earthquake and Tsunami, Japan. Road-impact data modified from MLIT, 2012, © OpenStreetMap contributors 2015. Distributed under a Creative Commons BY-SA License.**

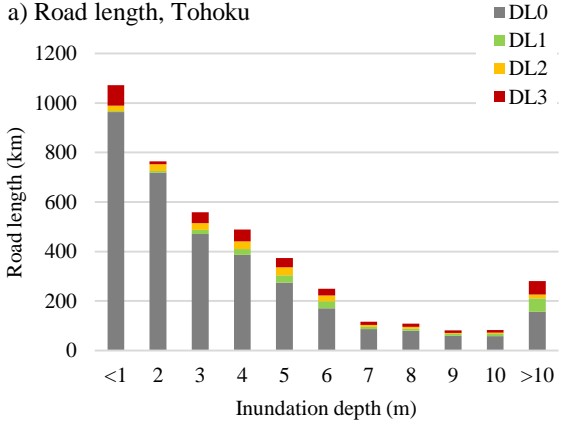

a) Road length, Tohoku

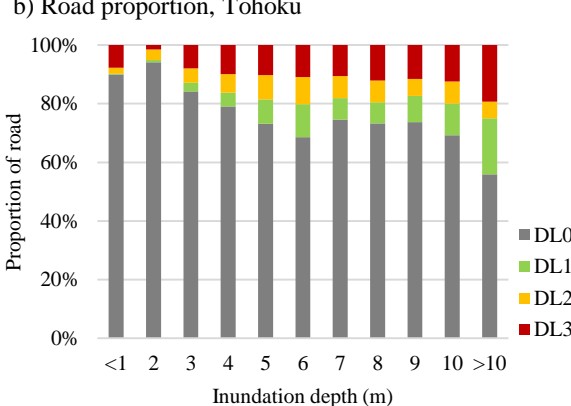

b) Road proportion, Tohoku

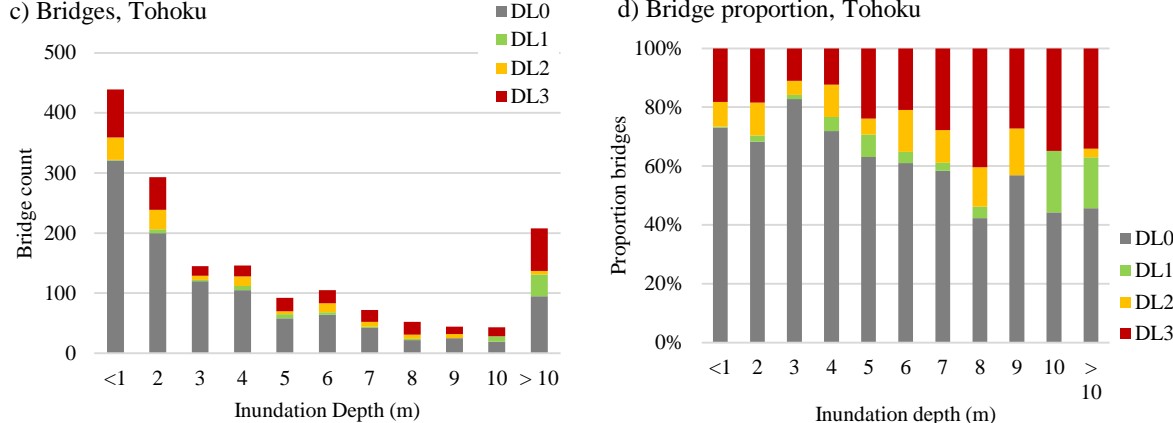

**Figure 5: Total length (a) and proportion (b) of exposed roads and number (c) and proportion (d) of exposed bridges, by inundation depth, in Miyagi and Iwate Prefectures for the 2011 Tohoku Earthquake and Tsunami**

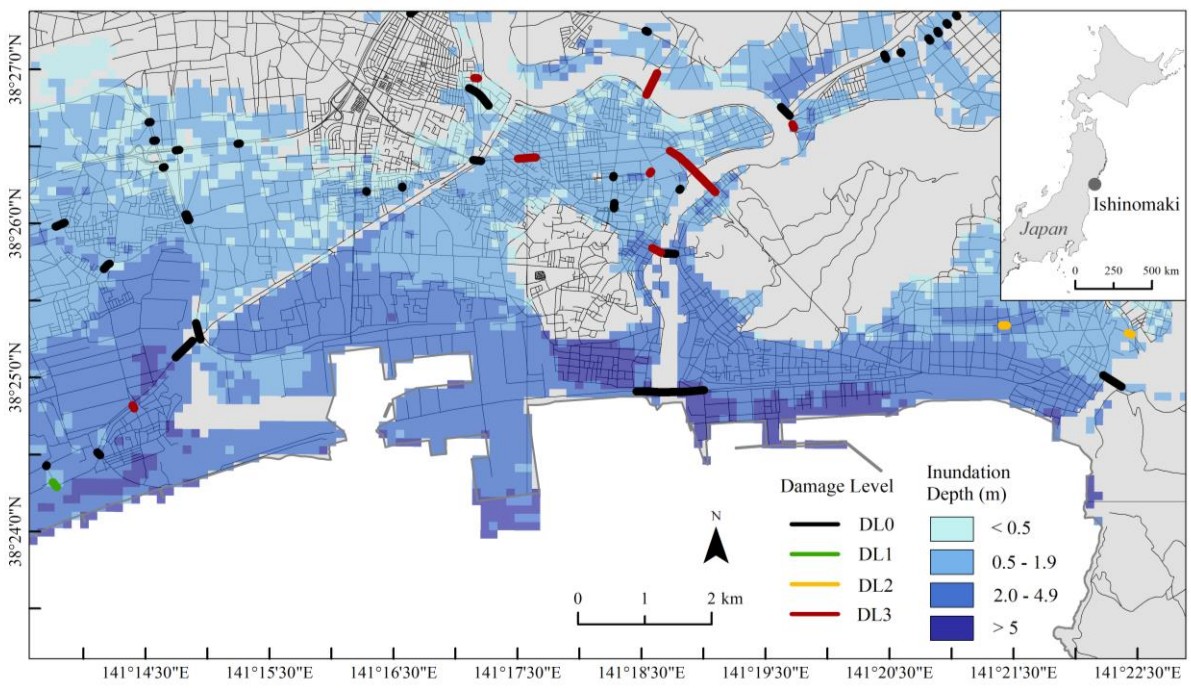

**Figure 6: Damage states for inundated bridges in Ishinomaki, Japan, for the 2011 Tohoku Earthquake and Tsunami. Bridge impact data modified from MLIT, 2012. © OpenStreetMap contributors 2015. Distributed under a Creative Commons BY-SA License.**

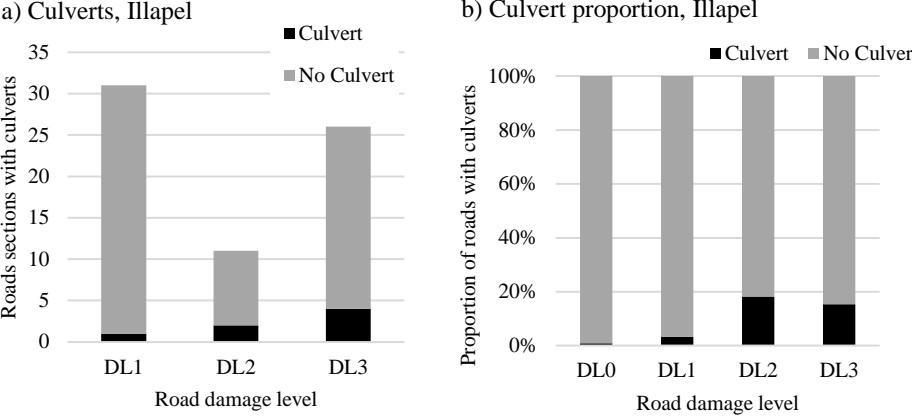

**Figure 7: Number (a) and proportion (b) of road sections with or without a culvert, by damage level, for the 2015 Illapel Earthquake and Tsunami. Note: DL0 had a count of 573 road sections (too many to represent in Figure 7a), with five having a culvert.**

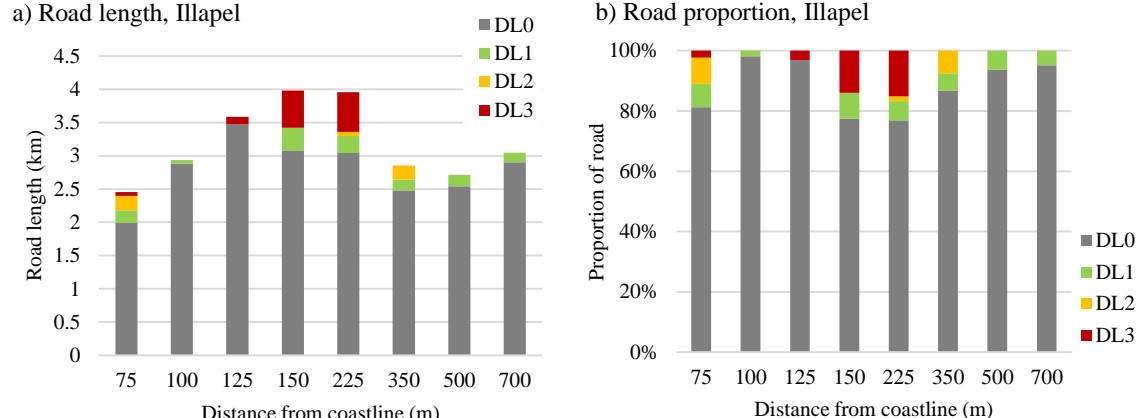

**Figure 8: Total length (a) and proportion (b) of exposed roads, by distance from the coastline (as a proxy for inundation energy), for the 2015 Illapel Earthquake and Tsunami**

**Table 2: Classification schema for road level of service for the 2015 Illapel Earthquake and Tsunami. Images taken as screenshots sourced from Puerto Creativo, 2015**

| Service Level | U | 0 | 1 | 2 | 3 |
|---|---|---|---|---|---|
| Service Level Description | Unknown (surface ponding) | No loss of service | Vehicle access at a reduced speed | All-wheel drive vehicle access at reduced speed | No vehicle access |
| Image | | | | | |

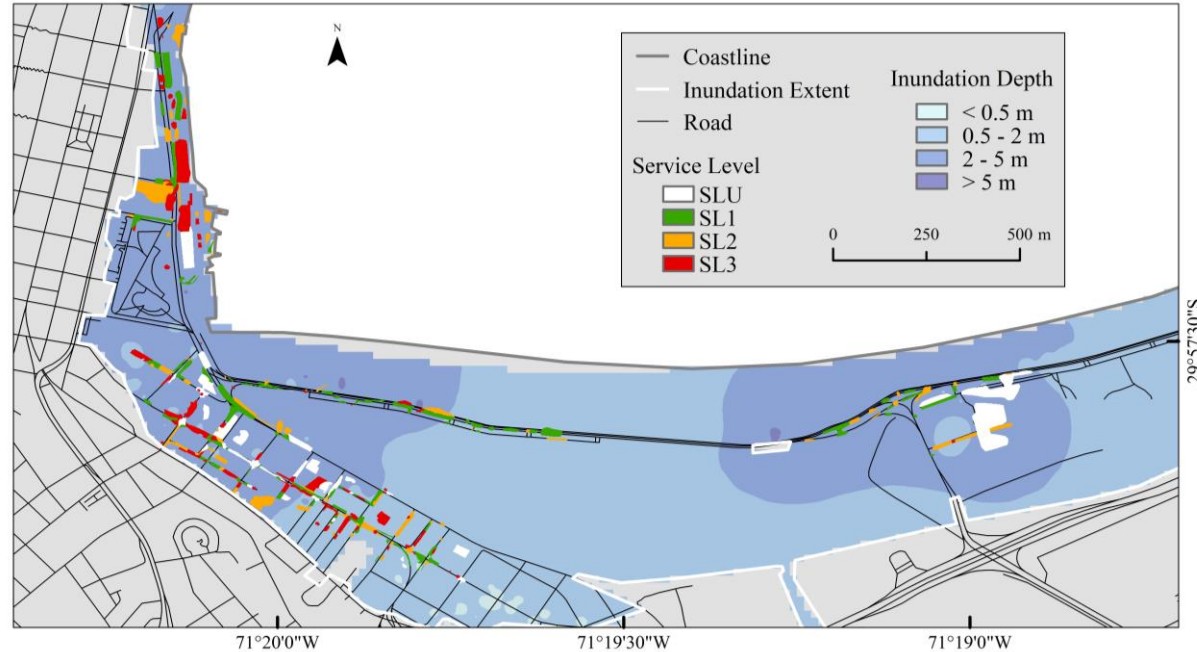

**Figure 9: Service levels associated with debris on roads in Coquimbo following the 2015 Illapel Earthquake and Tsunami, Chile, © OpenStreetMap contributors 2015. Distributed under a Creative Commons BY-SA License.**

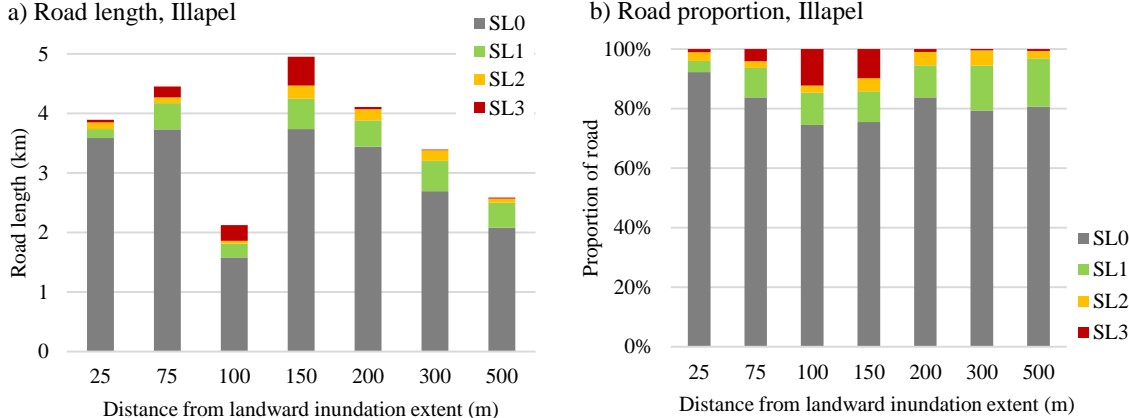

**Figure 10: Total length (a) and proportion (b) of exposed roads, considering levels of service, by distance from the landward inundation extent, for the 2015 Illapel Earthquake and Tsunami. Note SLU is not considered in analysis**

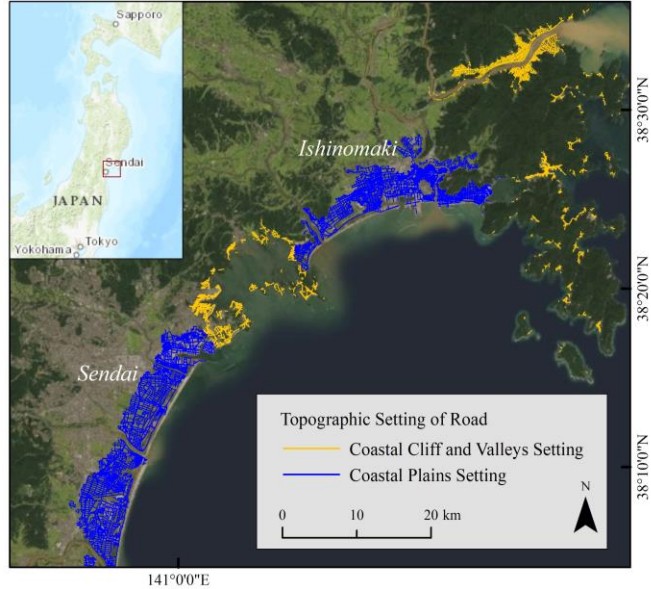

**Figure 11: Coastal topographic settings for inundated roads in Miyagi and Iwate Prefectures for the 2011 Tohoku Tsunami, Japan. Note: all roads North of Ishinomaki are Coastal Valleys; all roads South of Sendai are Coastal Plains. Road data modified from MLIT, 2012 and © OpenStreetMap contributors 2015. Distributed under a Creative Commons BY-SA License. Japan topographic imagery sourced from ESRI contributors, 2019a, Tohoku regional satellite imagery sourced from ESRI contributors, 2019b**

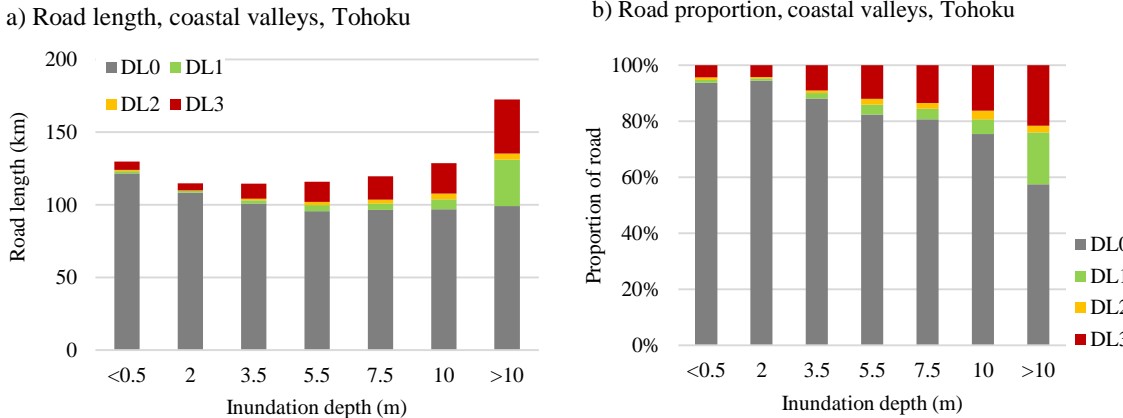

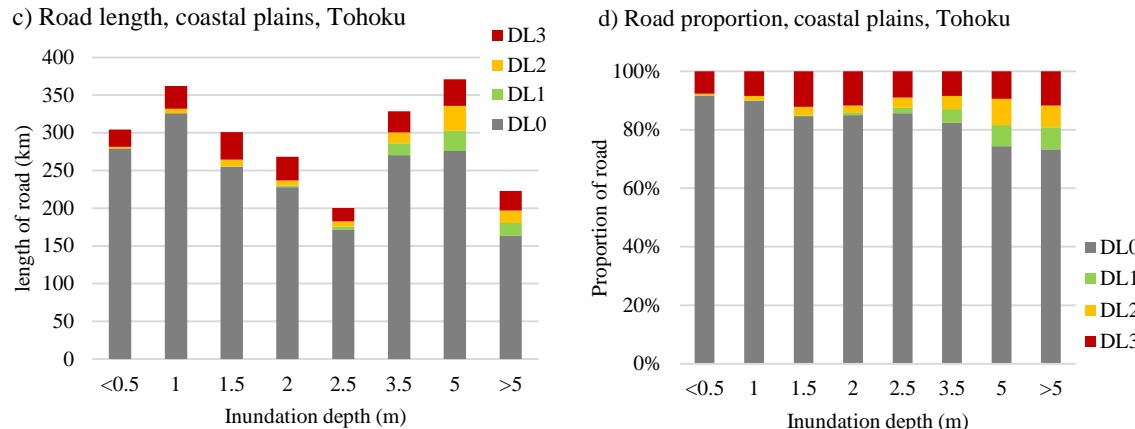

**Figure 12: Total length (a) and proportion (b) of exposed roads in a coastal valley topographic setting and total length (c) and proportion (d) of exposed roads in a coastal plain topographic setting, by inundation depth, for the 2011 Tohoku Earthquake and Tsunami**

**Table 3: Curve parameters for the tsunami fragility functions developed for transportation assets\***

| Fragility function | $\mu$ | $\sigma$ | $r^2$ |
|---|---|---|---|
| Tohoku ID MX Roads: DL1 | 3.33 | 2.51 | 0.83 |
| Tohoku ID MX Roads: DL2 | 5.31 | 3.77 | 0.82 |
| Tohoku ID MX Roads: DL3 | 5.76 | 3.18 | 0.80 |
| Tohoku ID MX Bridges: DL1 | 2.53 | 4.01 | 0.84 |
| Tohoku ID MX Bridges: DL2 | 5.52 | 8.00 | 0.55 |
| Tohoku ID MX Bridges: DL3 | 5.38 | 5.25 | 0.60 |
| Illapel DC MX Roads: DL1 | 0.16 | 0.00 | 0.34 |
| Illapel DC MX Roads: DL2 | 0.10 | 0.00 | 0.36 |
| Illapel DC MX Roads: DL3 | 0.07 | 0.00 | 0.31 |
| Illapel ID MX Roads: DL1 | 2.00 | 1.18 | 0.82 |
| Illapel ID MX Roads: DL2 | 2.47 | 1.23 | 0.71 |
| Illapel ID MX Roads: DL3 | 4.16 | 2.16 | 0.48 |
| Tohoku ID SR Road: DL1 | 3.68 | 1.64 | 0.82 |
| Tohoku ID SR Road: DL2 | 5.35 | 2.58 | 0.67 |
| Tohoku ID SR Road: DL3 | 6.04 | 2.77 | 0.52 |
| Tohoku ID LR Road: DL1 | 2.28 | 2.58 | 0.75 |
| Tohoku ID LR Road: DL2 | 3.21 | 4.06 | 0.65 |
| Tohoku ID LR Road: DL3 | 9.33 | 10.03 | 0.52 |
| Tohoku ID PR Road: DL1 | 2.22 | 2.68 | 0.75 |

| | | | |
|---|---|---|---|
| Tohoku ID PR Road: DL2 | 4.29 | 4.65 | 0.73 |
| Tohoku ID PR Road: DL3 | 4.77 | 3.29 | 0.53 |
| Tohoku ID MR Road: DL1 | 1.73 | 1.31 | 0.92 |
| Tohoku ID MR Road: DL2 | 2.23 | 1.75 | 0.90 |
| Tohoku ID MR Road: DL3 | 2.50 | 1.80 | 0.83 |
| Tohoku ID UR Road: DL3 | 0.83 | 4.99 | 0.76 |
| Tohoku CP ID MX Roads: DL1 | 4.88 | 4.07 | 0.90 |
| Tohoku CP ID MX Roads: DL2 | 8.25 | 6.74 | 0.87 |
| Tohoku CP ID MX Roads: DL3 | 17.01 | 12.42 | 0.73 |
| Tohoku CV ID MX Roads: DL1 | 3.40 | 1.75 | 0.94 |
| Tohoku CV ID MX Roads: DL2 | 5.07 | 3.02 | 0.93 |
| Tohoku CV ID MX Roads: DL3 | 5.42 | 3.11 | 0.95 |
| Tohoku ID CP SR Road: DL1 | 5.21 | 2.71 | 0.95 |
| Tohoku ID CP SR Road: DL2 | 5.15 | 2.58 | 0.90 |
| Tohoku ID CP SR Road: DL3 | 4.64 | 2.04 | 0.95 |
| Tohoku ID CV SR Road: DL1 | 3.03 | 1.11 | 0.58 |
| Tohoku ID CV SR Road: DL2 | 3.29 | 0.75 | 0.56 |
| Tohoku ID CV SR Road: DL3 | 3.33 | 0.63 | 0.58 |
| Tohoku ID CP LR Road: DL1 | 3.35 | 4.07 | 0.80 |
| Tohoku ID CP LR Road: DL2 | 3.49 | 4.17 | 0.81 |
| Tohoku ID CP LR Road: DL3 | 6.95 | 7.30 | 0.67 |
| Tohoku ID CV LR Road: DL1 | 1.33 | 3.43 | 0.56 |
| Tohoku ID CV LR Road: DL2 | 3.30 | 6.99 | 0.44 |
| Tohoku ID CV LR Road: DL3 | 16.31 | 16.31 | 0.339 |

*MX. = mixed construction, ID = Inundation depth as the HIM, DC = distance from coast as a HIM proxy, CP = coastal plains topography,*
*CV = coastal valley topography. SR = state road, LR = Main Local Road, PR = General Prefectural Road, MR = Municipalities Road, UR = Lowest class roads (unsealed), $r^2$ = Regression score.*

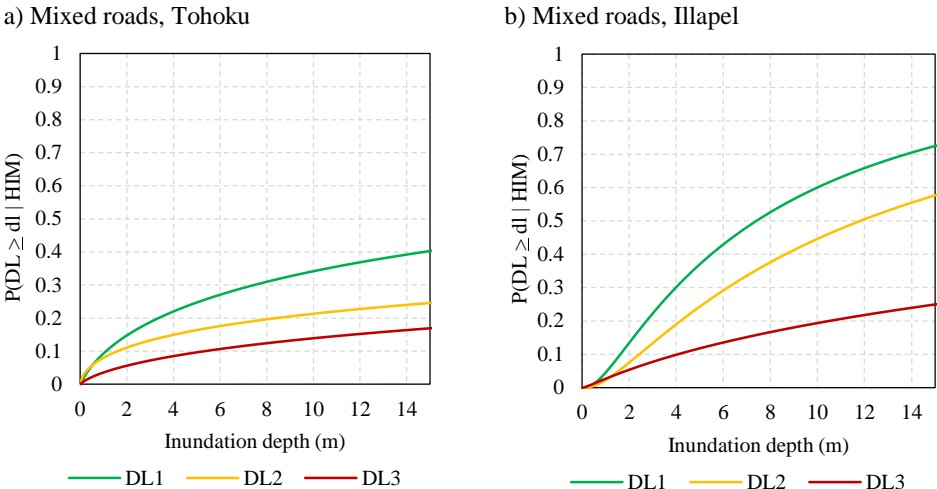

**Figure 13: Fragility functions for mixed construction roads (a) for the 2011 Tohoku Earthquake and Tsunami, Japan and for mixed construction roads (b) for the 2015 Illapel Earthquake and Tsunami, Chile**

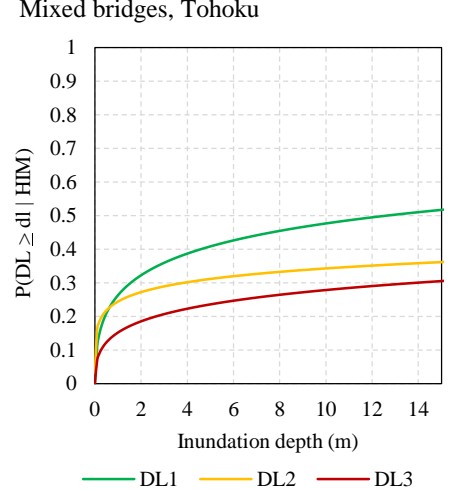

**Figure 14: Fragility function for mixed construction road bridges for the 2011 Tohoku Earthquake and Tsunami, Japan**

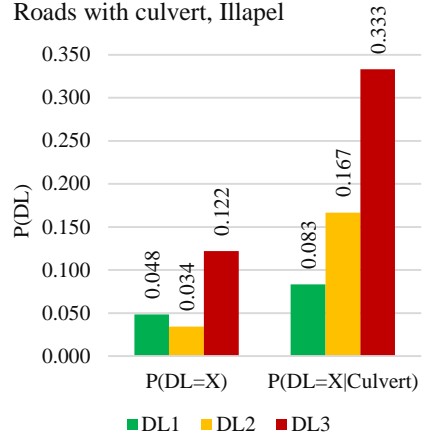

**Figure 15: Total road damage probability and increased total road damage probability with the presence of a culvert**

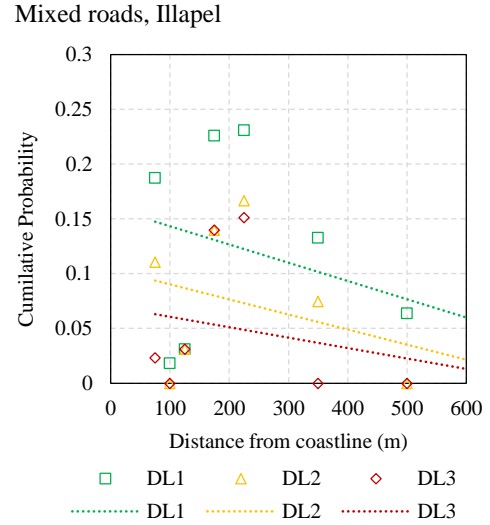

**Figure 16: Linear best fit probability of reaching or exceeding a given damage level, by distance from coastline (as a proxy for inundation energy, for the 2015 Illapel Tsunami, Coquimbo, Chile**

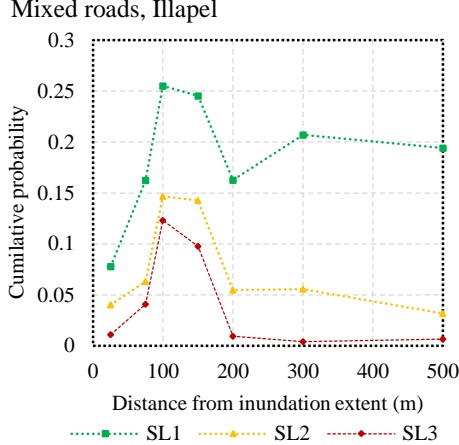

**Figure 17: Cumulative probability plot of road service levels compared with a distance from the in-land extent of tsunami inundation (as an indication of debris density sorting) for the 2015 Illapel Tsunami, Coquimbo, Chile.**

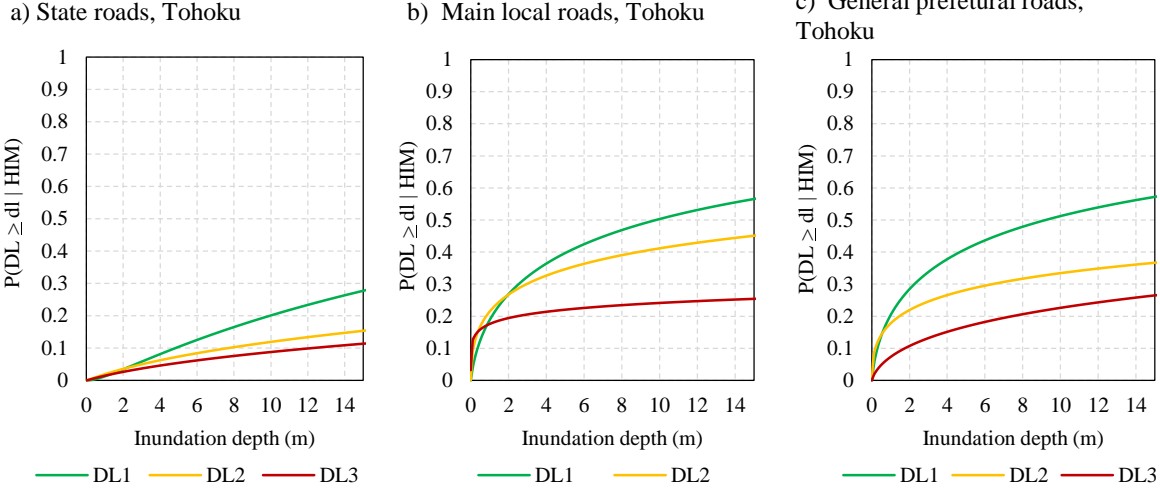

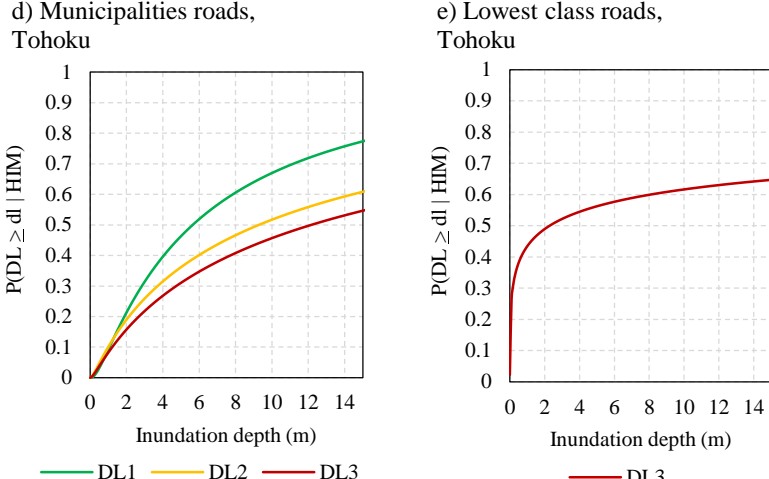

**Figure 18: Fragility functions for inundated State Roads (a), Main Local Roads (b), General Prefectural Roads (c), Municipalities Roads (d) and Lowest Class Roads (e) (as an indicator of construction type and materials) in Miyagi and Iwate prefectures following the 2011 Tohoku Earthquake and Tsunami, Japan.**

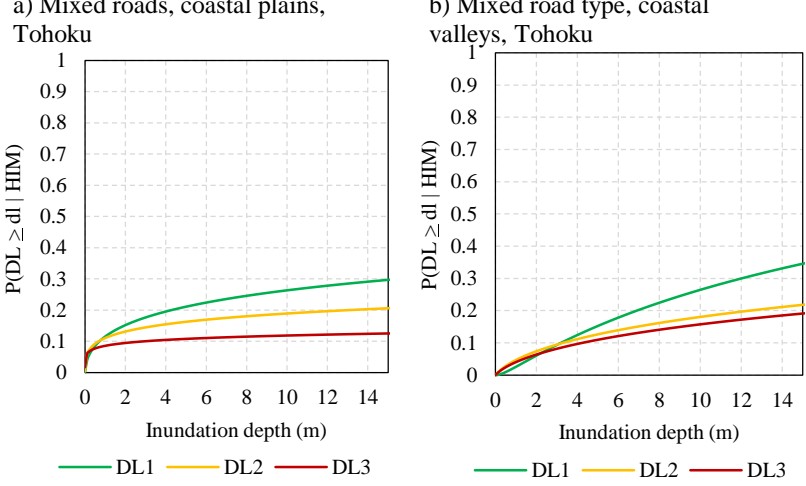

**Figure 19: Fragility functions for roads on coastal plains (a) and coastal valleys (b) for the 2011 Tohoku Earthquake and Tsunami, Japan**

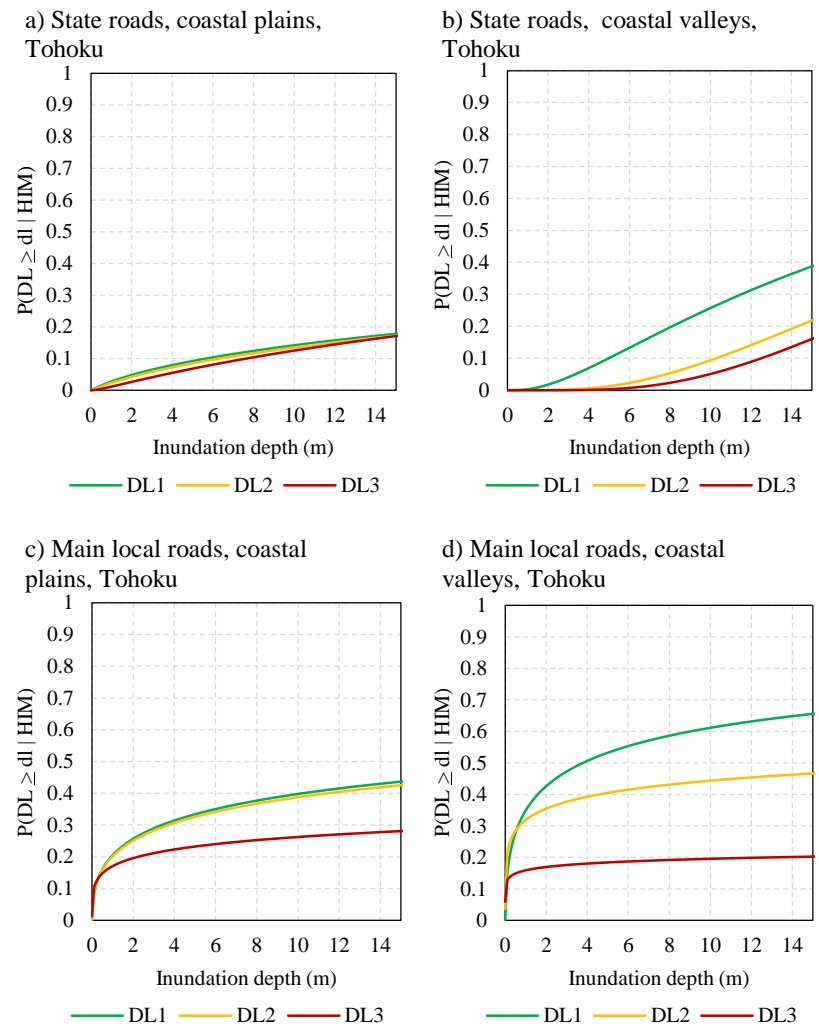

**Figure 20: Fragility functions for road State Roads on coastal plains (a) and coastal valleys (b), and for Main Local Roads on coastal plains (c) and coastal valleys (d) for the 2011 Tohoku Earthquake and Tsunami, Japan**