# Peer review of "Assessing Transportation Vulnerability to Tsunamis: Utilising Postevent Field Data from the 2011 Tohoku Tsunami, Japan, and the 2015 Illapel Tsunami, Chile"

_Natural Hazards and Earth System Sciences, 2019_

## Referee Comment (RC1) · Grant Wilson (Referee) · 30 Oct 2019

**Assessing Transportation Vulnerability to Tsunamis: Utilising Postevent Field Data from the 2011 Tohoku Tsunami, Japan, and the 2015 Illapel Tsunami, Chile**

This is an excellent piece of research and a great contribution to the scientific community. The paper is well written and lays out the problem and new research appropriately. I particularly like the inclusion of the limitations section so readers know when they can and cannot use the fragility functions.

Below are my minor comments and questions:

**Line 40:** add another ")" after the Koshimura et al., 2009 reference

**Line 49:** delete "field"

**Line 110:** change "infrastructural" to "infrastructure"

**Line 147:** delete extra ")" after the NZTA, 2014 reference

**Lines 169, 190 and Figure 4 and 6 captions:** it took me a while to realise that the area shown in Figure 4 and 6 is not the whole study area for the Tohoku event. Particularly when I was comparing the data in Figure 5a which shows thousands of bridges and Figure 6 which shows 20+ bridges. I would suggest at line 169 and line 190 in new sentences state the Figure 4 (Figure6) show an example of observes damage levels for roads (bridges) in the town of Ishinomaki within the study area. Or incorporate something similar into the Figure 4 and 6 captions.

**Line 208 and Figure 7 caption:** incorporate a similar phrase "DL0 had a count of 573 road sections (too many to represent in Figure 7a), with five having a culvert." into the caption of Figure 7.

**Lines 214-215:** "This was observed for road assets in Coquimbo as damage levels reduced with distance from the coast." Is this the only reason for this trend? Could it be caused by other effects such as change in inundation depth, topography, change in road construction?

**Line 218:** add "s" after "function"

**Line 219:** In terms of the debris based level of service, was any consideration given to areas that, pre-tsunami, might have had a higher concentration of maternal that could become debris e.g. construction sites, industrial area, etc? These areas might have higher density of debris post-tsunami compared with the method you used.

**Line 222:** change "assets" to "asset's"

**Line 225:** change "levels of service" to "service levels (SL)"

**Lines 227-229:** the sentence starting with "To account for potential horizontal…" seems to be missing some words before or after the brackets. I don't know what is trying to be said to offer a re-write.

**Line 247:** spell out "CDF"

**Line 282:** delete "even"

**Line 286:** Start a new sentence after "(Figure 13b)"

**Lines 298-299:** Can you explain why these did not warrant the development of fragility functions?

**Line 326:** delete "also"

**Line 337:** "at around 2m" on Figure 19a it looks a lot less than 2m, maybe 0.2m?

**Lines 338-339:** the sentence starting with "However, the least squares regression…" is this correct? Looking at Table 3 it seems the coastal plains r2 values are lower than the valleys.

**Line 348:** add references for "previous studies"

**Line 405:** change "behave" to "have"

**Line 407:** change "may a" to "may **be** a"

---

## Referee Comment (RC2) · Constance Ting Chua (Referee) · 15 Nov 2019

The paper provides an interesting insight into the impacts of tsunami on roads and bridges, and I do agree with the authors that it is worthwhile to look at tsunami impacts on critical infrastructure. The language used in this paper is clear, albeit with some grammatical errors. The field survey conducted in this study has included quite an interesting and extensive suite of observations, and was conducted in consideration of the Tohoku post-event survey dataset. That said, I still have some concerns about the paper.

[Figure]

Throughout this paper, it appears that the differences in road use type, construction type, coastal topography in influencing damage are very much limited to only one dataset (either the Tohoku or the Illapel dataset) in the analysis. While it is not particularly an issue for me, it does contradict with what the authors set out in the problem statement in page 2, line 44-45. The analysis still does seem to describe about local characteristics. My suggestion is to show a table which summarises the dataset which was used to examine the influence of each factor (e.g. culvert, distance from coast).

I am also confused about the development of fragility functions for the different types of analysis (e.g. influence of distance from coastline, coastal topography, debris, road use type). The authors seem to have developed fragility functions for some, and not for the others. I assumed from the objectives outlines in the abstract that fragility functions would be developed for different factors. My suggestion would be to separate the factors for which the authors have developed fragility functions for, and those which they have only conducted spatial analysis (e.g. distribution of debris etc) for into separate sections.

Specific comments for the paper are included in the attached file. I hope that the authors would address them and I do hope that with the revised version, this paper would prove to be a worthy pioneer work for future studies on tsunami damage to transportation network.

Please also note the supplement to this comment:
https://www.nat-hazards-earth-syst-sci-discuss.net/nhess-2019-332/nhess-2019-332-RC2-supplement.pdf

**Supplement:**

Specific comments

**Grammar:** Please make sure that the grammar is consistent throughout the paper. It can be either "all was" or "All were", "Data was" or "Data were", etc. Make sure it's the same throughout the paper.

**Abstract**: Perhaps the authors could provide in the abstract a list of damage characteristics which were examined in this study, i.e. influence of culverts, influence of inundation distance, influence of debris, influence of road use type, influence of topography. These were the main themes which were explored throughout the paper but were not explicitly stated in the abstract and introduction.

**Page 2, Line 44 - 45:** "Fragility functions derived from a single tsunami event means they will be characteristic of local asset and event characteristics". From this statement, I assumed that the authors were going to develop fragility functions based on collective data from both events (i.e. combining two sets of data to create a single function).

**Page 2, Line 48**: I am not entirely convinced that there is a strong correlation between inundation depth and impact, perhaps for higher levels of damage, yes (it is a very broad statement to make). Rather than risking it, I would suggest that the authors look for literature that supports this statement.

**Page 3, Line 78 – 79**: Please check if "tsunami waves exceeding 30 m in inundation depth" is an accurate description. Having cross-checked with the referenced paper (Kazama and Noda, 2012), it seems to me that they are referring to inundation heights. Please be aware that they are inherently different terms. Inundation height usually refers to height of inundation above MSL, and inundation depth refers to the depth of inundation above ground level. Please be careful, and make sure the measurements which were used in this paper for analysis are referring to the same unit, i.e. the data collected in Illapel were indeed Inundation Depths and the HIM taken from the MLIT database is indeed inundation depths and not height. As far as I am aware, the MLIT database usually provides a number of measurements for inundation.

**Page 4, Line 120 -121**: "Areas with flat topography are not typically consistent with direct road damage from shaking alone. However, where soil liquefaction occurred, then this could have resulted in damage." I do not really understand what the first sentence meant, it could be better phrased.

**Page 6, Line 160:** Just a suggestion, because I am not sure how best to structure the methodology section. Before talking about splitting the data into inundation depth bins, the authors can perhaps first mention how they would derive the fragility functions and that the data would be split into bins when performing their statistical analysis. It is just a suggestion.

**Page 6, Line 160 – 170:** The data for Illapel was split into inundation depth bins of 0.25m and 1 m for Tohoku dataset. Why is this?

**Page 8, Line 234 - 235**: "There was no such empirical source of debris density observations available for the 2011 Tohoku tsunami, so this is not considered in the analysis". I am not certain if the authors meant that debris density is not considered in this study or just the Tohoku dataset is excluded.

**Page 10, Line 298-299:** Why did the analysis for distance from coast not warrant the development of fragility functions?

**Page 10, Line 304:** Be careful here, tsunami debris transport is also a function of velocity. I suggest the authors refer to Charvet et al. (2014) Section 2.2 (pg 1855) to understand more about the flow characteristics which influence the different types of forces acting on structures.

**Page 10, Section 3.3:** I just want to confirm with the authors if debris density refers to the size of the debris or the distribution of debris. Not a major issue but why was debris distribution measured from inland inundation extent, instead of the coastline? I would assume that it is easier to imagine how distribution differs as we move landwards, e.g. higher distribution of debris nearer to shore and lower distribution away from the shore?

**Page 15, Line 462 – 464:** I am not sure what "until maximum inundation depths are exceeded" meant, how are they exceeded?

Also, conclusions in this bullet point seem to contradict conclusions of other paper. Authors mentioned that coastal valleys result in higher inundation depths and lower velocities. Please refer to Suppasri et al. (2015) page 585 - "for a given inundation depth, a higher damage probability exists on the ria coast due to higher flow velocity". I just wondered if perhaps the dataset used is from Ishinomaki (which fringes a ria coast but does not entirely lie in a ria coast).

Charvet, I., A. Suppasri, and F. Imamura. "Empirical fragility analysis of building damage caused by the 2011 Great East Japan tsunami in Ishinomaki city using ordinal regression, and influence of key geographical features." *Stochastic environmental research and risk assessment* 28.7 (2014): 1853-1867.

Suppasri, Anawat, et al. "Fragility curves based on data from the 2011 Tohoku-Oki Tsunami in Ishinomaki city, with discussion of parameters influencing building damage." *Earthquake Spectra* 31.2 (2015): 841-868.

---

## Short Comment (SC1) · 19 Nov 2019

This paper proposed fragility functions for transportation assets. One of the study areas is Coquimbo, Chile. Even though there is no studies related to transportation vulnerability to tsunami in this area, there are other published papers that analyze vulnerability after the 2015 Coquimbo tsunami. See for example Izquierdo et al (2018) "Analysis and validation of the PTVA tsunami building vulnerability model using the 2015 Chile post-tsunami damage data in Coquimbo and La Serena cities" and Aránguiz et al (2018) "Development and application of a tsunami fragility curve of the 2015

tsunami in Coquimbo, Chile". It would be interesting to discuss some results of those papers in the results and discussion sections. For example, in the latter paper, we compared different curves from other places and made comments on ria and plain coasts. In addition, we made comments about the effects of the wetland and proximity to the coast on damage to houses.

—————————————————

---

## Author Comment (AC1) · 7 Jan 2020

Thank you for your comment. it was an oversight that these studies where not already cited in the manuscript. We have now added reference to the coastal setting, noted in Aránguiz et al (2018), into the discussion (Lines 400 – 401). We have also added comparison with building damage and distance to the coastline in Lines 312-314. We have also added these references to the relevant sections of the introduction.

[Figure]

2019-332, 2019.

---

## Author Comment (AC2) · 7 Jan 2020

**Responses to Reviewer #1 comments:**

*Thank you for the critique of this manuscript, it is much appreciated. We have responded to all of your comments below. Responses are in bold Italics. All line numbers in the responses relate to the updated document (once the revised manuscript is prepared).*

This is an excellent piece of research and a great contribution to the scientific community. The paper is well written and lays out the problem and new research appropriately. I particularly like the inclusion of the limitations section so readers know when they can and cannot use the fragility functions.

Below are my minor comments and questions:

Line 40: add another ")" after the Koshimura et al., 2009 reference

*Line 44: Removed "(" before the reference*

Line 49: delete "field"

*Line 53: removed "field"*

Line 110: change "infrastructural" to "infrastructure"

*Line 115: changed "infrastructural" to "infrastructure"*

Line 147: delete extra ")" after the NZTA, 2014 reference

*Line 152: deleted ")"*

Lines 169, 190 and Figure 4 and 6 captions: it took me a while to realise that the area shown in Figure 4 and 6 is not the whole study area for the Tohoku event. Particularly when I was comparing the data in Figure 5a which shows thousands of bridges and Figure 6 which shows 20+ bridges. I would suggest at line 169 and line 190 in new sentences state the Figure 4 (Figure6) show an example of observes damage levels for roads (bridges) in the town of Ishinomaki within the study area. Or incorporate something similar into the Figure 4 and 6 captions.

*Lines 174-175: "(Figure 4)." changed to "Figure 4 shows an example of observed damage levels for roads in the town of Ishinomaki within the study area."*

*Lines 197 - 198: "(Figure 6)." changed to "Figure 6 shows an example of observed damage levels for bridges in the town of Ishinomaki within the study area."*

Line 208 and Figure 7 caption: incorporate a similar phrase "DL0 had a count of 573 road sections (too many to represent in Figure 7a), with five having a culvert." into the caption of Figure 7.

*Figure 7 caption: "Note: DL0 had a count of 573 road sections (too many to represent in Figure 7a), with five having a culvert." Added.*

Lines 214-215: "This was observed for road assets in Coquimbo as damage levels reduced with distance from the coast." Is this the only reason for this trend? Could it be caused by other effects such as change in inundation depth, topography, change in road construction?

*We agree with your criticism; however, we do point out in lines 212-215 that distance from a coastline is not directly linked to observed damage, but rather the deteriorating wave energy and, therefore, hazard intensity. This is the reason we do not develop*

***any kind of vulnerability function for 'distance from coastline'. We have added more description to lines 225-226 around this: "Since distance from the coastline is not a direct impact causing process, therefore, the analysis is not conducive with fragility functions, so none are developed". We also acknowledge this as a limitation on Lines: 398-401 "Whereas the more localised data of Illapel has more consistent quantities of data across the range of inundation depths but is also limited by the overall data size. For example, the dataset did not warrant the comparison of different coastal settings since only flat topography was represented in the study area, which is also noted by Aránguiz et al., 2018 in the context of building vulnerability."***

Line 218: add "s" after "function"

***Line 226: "s" added***

Line 219: In terms of the debris based level of service, was any consideration given to areas that, pre-tsunami, might have had a higher concentration of maternal that could become debris e.g. construction sites, industrial area, etc? These areas might have higher density of debris post-tsunami compared with the method you used.

***Although site-specific debris origin is not specifically considered, as that is beyond the scope of this work, the authors are aware of this. More context has been added to Lines 238-240 in consideration of this point. "The local sea port, of which are typically well-defined regions of debris origin (Naito et al., 2014), was located along the South-West to North-West inundated coastline."***

Line 222: change "assets" to "asset's"

***Line 230: apostrophe added***

Line 225: change "levels of service" to "service levels (SL)"

***Line 233: changed "levels of service (Table 2)" to "service levels (SL), as defined in Error! Reference source not found."***

Lines 227-229: the sentence starting with "To account for potential horizontal…" seems to be missing some words before or after the brackets. I don't know what is trying to be said to offer a rewrite.

***Thank you, this was an oversight, "was used" has been added to Line 237***

Line 247: spell out "CDF"

***Line 257: "CDF" changed to "cumulative distribution function"***

Line 282: delete "even"

***Line 292: removed "even"***

Line 286: Start a new sentence after "(Figure 13b)"

***Line 296: full stop added***

Lines 298-299: Can you explain why these did not warrant the development of fragility functions?

***It is because distance from a coastline is not a hazard intensity  as such and is not the process resulting in damage directly. We have added more clarity around this: "(Sub-section 2.2.2)" moved to the end of sentence (Line 209) and more explanation is***

*now provided in subsection 2.2.2 (Lines 225-226). More content around the small sample sized used is also added to line 463-465 "particularly as the small sample size at Coquimbo reduces the ability to derive a robust statistical sample. Therefore, the observation remains qualitative and the parameters require further investigation from future events"*

Line 326: delete "also"

*Line 339: deleted "also"*

Line 337: "at around 2m" on Figure 19a it looks a lot less than 2m, maybe 0.2m?

*Thank you, this was an oversight. "~0.08" changed to "~0.09" (Line 348), "2m" is changed to "0.03 m" (Line 350), and "4m" is changed to "3m" (Line 351)*

Lines 338-339: the sentence starting with "However, the least squares regression…" is this correct? Looking at Table 3 it seems the coastal plains r2 values are lower than the valleys.

*Thank you, you are correct about the r2 values being lower for coastal plains for the mixed construction roads, this was an oversight. The sentence "However, the $r^2$ values for coastal valleys are particularly low, so the comparisons between each coastal setting my not be entirely representative of true vulnerability" is meant for the next paragraph regarding varied construction type and the influence on vulnerability in each topographic setting. It has been moved to lines 359-360 and "However, the $r^2$ values for coastal plains are lower than coastal valleys and, therefore, may not be as representative of true vulnerability" has been added to lines 351-352.*

Line 348: add references for "previous studies"

*Line 363: reference added*

Line 405: change "behave" to "have"

*Line 406: changed "behave" to "have"*

Line 420: change "may a" to "may be a"

*Line 422: changed "may a" to "may be a"*

---

## Author Comment (AC3) · 7 Jan 2020

**Responses to Reviewer #2 comments:**

***Thank you for your detailed review of this manuscript. We have responded to all of your comments below. Responses are in bold italics. All line numbers refer to the updated manuscript document.***

The paper provides an interesting insight into the impacts of tsunami on roads and bridges, and I do agree with the authors that it is worthwhile to look at tsunami impacts on critical infrastructure. The language used in this paper is clear, albeit with some grammatical errors. The field survey conducted in this study has included quite an interesting and extensive suite of observations, and was conducted in consideration of the Tohoku post-event survey dataset. That said, I still have some concerns about the paper.

Throughout this paper, it appears that the differences in road use type, construction type, coastal topography in influencing damage are very much limited to only one dataset (either the Tohoku or the Illapel dataset) in the analysis. While it is not particularly an issue for me, it does contradict with what the authors set out in the problem statement in page 2, line 44-45. The analysis still does seem to describe about local characteristics. My suggestion is to show a table which summarises the dataset which was used to examine the influence of each factor (e.g. culvert, distance from coast).

I am also confused about the development of fragility functions for the different types of analysis (e.g. influence of distance from coastline, coastal topography, debris, road use type). The authors seem to have developed fragility functions for some, and not for the others. I assumed from the objectives outlines in the abstract that fragility functions would be developed for different factors. My suggestion would be to separate the factors for which the authors have developed fragility functions for, and those which they have only conducted spatial analysis (e.g. distribution of debris etc) for, into separate sections. Specific comments for the paper are included in the attached file. I hope that the authors would address them and I do hope that with the revised version, this paper would prove to be a worthy pioneer work for future studies on tsunami damage to transportation network.

***As recommended, we have added new content to the abstract (lines 16-19) to clarify the difference between observations and fragility function development for each of the influencing factors mentioned in the study. We acknowledge your concern with this further in the manuscript; however, we believe it is clearly stated throughout the manuscript when fragility functions are, and are not, developed. We also believe Table 3 does an adequate job of summarising the influencing factors considered for each fragility function, and the corresponding datasets. We also believe the limitations of each data-set are adequately mentioned throughout, and more specifically in sub-section 4.1.***

Specific Coments:

Grammar: Please make sure that the grammar is consistent throughout the paper. It can be either "all was" or "All were", "Data was" or "Data were", etc. Make sure it's the same throughout the paper.

***Thank you these have been corrected throughout the text where relevant.***

Abstract: Perhaps the authors could provide in the abstract a list of damage characteristics which were examined in this study, i.e. influence of culverts, influence of inundation distance, influence of debris, influence of road use type, influence of topography. These were the main

themes which were explored throughout the paper but were not explicitly stated in the abstract and introduction.

***As recommended, we have added new content to the abstract (lines 16-19) to clarify the difference between observations and fragility function development for each of the influencing factors mentioned in the study***

Page 2, Line 44 - 45: "Fragility functions derived from a single tsunami event means they will be characteristic of local asset and event characteristics". From this statement, I assumed that the authors were going to develop fragility functions based on collective data from both events (i.e. combining two sets of data to create a single function).

***This statement supports the development of fragility functions from a range of events covering different event parameters, asset types and intensity measures. We do not think combining these datasets for a single fragility function is practical given there are only two data sets, which are very different in terms of data quantity and quality. They are also very different in terms of asset and event characteristics. We believe this point is adequately referred to throughout the manuscript, particularly in all subsections of sections 2 and 3 where each parameter is given an overview of data quantity and quality.***

Page 2, Line 48: I am not entirely convinced that there is a strong correlation between inundation depth and impact, perhaps for higher levels of damage, yes (it is a very broad statement to make). Rather than risking it, I would suggest that the authors look for literature that supports this statement.

***This is covered in the Introduction (Lines 40-55, e.g. "However tsunami hazard and impact studies to date are almost unanimous in that no single HIM can fully encapsulate the characteristics of tsunami impacts (Bojorquez et al., 2012; Gehl and D'Ayala, 2015; Macabuag et al., 2017; Sousa et al., 2014)").***

Page 3, Line 78 – 79: Please check if "tsunami waves exceeding 30 m in inundation depth" is an accurate description. Having cross-checked with the referenced paper (Kazama and Noda, 2012), it seems to me that they are referring to inundation heights. Please be aware that they are inherently different terms. Inundation height usually refers to height of inundation above MSL, and inundation depth refers to the depth of inundation above ground level. Please be careful, and make sure the measurements which were used in this paper for analysis are referring to the same unit, i.e. the data collected in Illapel were indeed Inundation Depths and the HIM taken from the MLIT database is indeed inundation depths and not height. As far as I am aware, the MLIT database usually provides a number of measurements for inundation.

***The statement "tsunami waves exceeding 30 m inundation depth" is consistent with the hazard intensity data we are using. To clarify that this was only at some isolated extremes, we have added "in some locations" (Line 25), and "in some extreme cases" (Line 84). Line 85: We have removed one of the references to avoid confusion between 'height' and 'depth' terminologies. Thank you for highlighting this oversight. We have kept the reference to the MLIT dataset (Line 85).***

Page 4, Line 120 -121: "Areas with flat topography are not typically consistent with direct road damage from shaking alone. However, where soil liquefaction occurred, then this could have resulted in damage." I do not really understand what the first sentence meant, it could be better phrased.

***This sentence is acknowledging a limitation in the data. We have added "which is not accounted for in this study" to clarify (Line 126)***

Page 6, Line 160: Just a suggestion, because I am not sure how best to structure the methodology section. Before talking about splitting the data into inundation depth bins, the authors can perhaps first mention how they would derive the fragility functions and that the data would be split into bins when performing their statistical analysis. It is just a suggestion.

***Thank you, but we respectfully disagree and have decided to keep the current paragraph structure because it reflects the workflow used. Lines 162-163 do mention the aim is to develop fragility functions ("The most common HIM is inundation depth, and the first step was to use this data to calculate fragility functions for mixed construction assets")***

Page 6, Line 160 – 170: The data for Illapel was split into inundation depth bins of 0.25m and 1 m for Tohoku dataset. Why is this?

***This was done since there were lower hazard intensities in Coquimbo (relative to the Tohoku dataset). A new sentence has been added to make this more clear (Lines 176-177 "Larger inundation depth bins were used compared with the Illapel dataset (i.e. 1m vs 0.25m), as there were greater hazard intensity values (> 10 m vs < 4 m)")***

Page 8, Line 234 - 235: "There was no such empirical source of debris density observations available for the 2011 Tohoku tsunami, so this is not considered in the analysis". I am not certain if the authors meant that debris density is not considered in this study or just the Tohoku dataset is excluded.

***Thank you, we have added "of the Tohoku dataset" to the end of the sentence (line 246) to make this more clear.***

Page 10, Line 298-299: Why did the analysis for distance from coast not warrant the development of fragility functions?

***It is because distance from a coastline is not a hazard intensity as such and is not the process resulting in damage directly. We have added more clarity around this: "(Sub-section 2.2.2)" moved to the end of sentence (Line 209) and more explanation is now provided in subsection 2.2.2 (Lines 225-226). More content around the small sample sized used is also added to line 463-465 "particularly as the small sample size at Coquimbo reduces the ability to derive a robust statistical sample. Therefore, the observation remains qualitative and the parameters require further investigation from future events"***

Page 10, Line 304: Be careful here, tsunami debris transport is also a function of velocity. I suggest the authors refer to Charvet et al. (2014) Section 2.2 (pg 1855) to understand more about the flow characteristics, which influence the different types of forces acting on structures.

***Thank you, this was an oversight. We have added ", inundation velocity" to line 316 and included Charvet et al. (2014) in the citation (line 317).***

Page 10, Section 3.3: I just want to confirm with the authors if debris density refers to the size of the debris or the distribution of debris. Not a major issue but why was debris distribution measured from inland inundation extent, instead of the coastline? I would assume that it is easier to imagine how distribution differs as we move landwards, e.g. higher distribution of debris nearer to shore and lower distribution away from the shore?

*We are referring to both size and distribution, but in the context of vehicle accessibility (i.e. a single large object would be classified the same as many small objects if both had the same effect on accessibility). We use distance from the inundation extent because we have identified the exact opposite of what you are suggesting (specifically in regards to reduced service level on roads).*

Page 15, Line 462 – 464: I am not sure what "until maximum inundation depths are exceeded" meant, how are they exceeded?

*We were referring to coastal valleys typically generating higher inundation depths (compared to plains) due to the topography restricting flow and allowing the waves to slow and increase in height. As with the below response this has been re-worded to avoid confusion (Lines 479-481).*

Also, conclusions in this bullet point seem to contradict conclusions of other paper. Authors mentioned that coastal valleys result in higher inundation depths and lower velocities. Please refer to Suppasri et al. (2015) page 585 - "for a given inundation depth, a higher damage probability exists on the ria coast due to higher flow velocity". I just wondered if perhaps the dataset used is from Ishinomaki (which fringes a ria coast but does not entirely lie in a ria coast).

*Thank you for pointing this out. Stating that there were higher velocities on coastal plains was an oversight, and we have re-worded the aforementioned bullet point (Lines 479-481) to reflect this, while still acknowledging that in some cases the fragility functions show higher vulnerability on coastal plains at low inundation depths. We have also added more clarity to this in the abstract to avoid confusion (Line 24-25)*

Charvet, I., A. Suppasri, and F. Imamura. "Empirical fragility analysis of building damage caused by the 2011 Great East Japan tsunami in Ishinomaki city using ordinal regression, and influence of key geographical features." Stochastic environmental research and risk assessment 28.7 (2014): 18531867.

Suppasri, Anawat, et al. "Fragility curves based on data from the 2011 Tohoku-Oki Tsunami in Ishinomaki city, with discussion of parameters influencing building damage." Earthquake Spectra 31.2 (2015): 841-868.